# Combined short-term and long-term emission controls improve air quality sustainably in China

Zhang Wen [1,2,9], Xin Ma[1,9], Wen Xu [1,9], Ruotong Si[1], Lei Liu [1,3], Mingrui Ma[4], Yuanhong Zhao[5], Aohan Tang[1], Yangyang Zhang[1], Kai Wang[1], Ying Zhang[1], Jianlin Shen [6], Lin Zhang [7], Yu Zhao [4], Fusuo Zhang [1], Keith Goulding [8] & Xuejun Liu [1] ✉

The effectiveness of national policies for air pollution control has been demonstrated, but the relative effectiveness of short-term emission reduction measures in comparison with national policies has not. Here we show that short-term abatement measures during important international events substantially reduced $PM_{2.5}$ concentrations, but air quality rebounded to pre-event levels after the measures ceased. Long-term adherence to strict emission reduction policies led to successful decreases of 54% in $PM_{2.5}$ concentrations in Beijing, and 23% in atmospheric nitrogen deposition in China from 2012 to 2020. Incentivized by "blue skies" type campaigns, economic development and reactive nitrogen pollution are quickly decoupled, showing that a combination of inspiring but aggressive short-term measures and effective but durable long-term policies delivers sustainable air quality improvement. However, increased ammonia concentrations, transboundary pollutant flows, and the complexity to achieving reduction targets under climate change scenarios, underscore the need for the synergistic control of multiple pollutants and inter-regional action.

China's economy has rapidly grown over the past 40 years. At the same time, environmental issues have become increasingly severe, damaging ecosystems and human health[1,2]. People's experience of air pollution is widespread, direct, and intense. Consequently, there is a strong demand from the public for improved air quality. Starting in the 1980s, China began to focus on the damage and necessary control measures of sulfur dioxide ($SO_2$) emissions and acid rain[3]. After 2010, attention to air quality improvement shifted to reactive nitrogen (Nr)

pollution and fine particulate matter ($PM_{2.5}$)[4,5]. During this period, as the economy steadily developed, some important international mega-events were organized in China. To ensure "blue skies" (i.e. clean air) the Chinese government implemented many short-term pollutant emission control measures during the events (each lasting for about a month), including the suspension of industrial activities to decrease $SO_2$ and nitrogen oxides ($NO_X$) emissions, and the reduction of vehicle movements in and around the event locations mainly focused on $NO_X$,

[1]State Key Laboratory of Nutrient Use and Management, College of Resources and Environmental Sciences, National Academy of Agriculture Green Development, China Agricultural University, Beijing 100193, China. [2]State Environmental Protection Key Laboratory of Environmental Pollution and Greenhouse Gases Co-control, Chinese Academy of Environmental Planning, Beijing 100041, China. [3]College of Earth and Environmental Sciences, Lanzhou University, Lanzhou 730000, China. [4]State Key Laboratory of Pollution Control & Resource Reuse and School of Environment, Nanjing University, Nanjing 210008, China. [5]College of Oceanic and Atmospheric Sciences, Ocean University of China, Qingdao 266100, China. [6]Instute of Subtropical Agriculture, Chinese Academy of Sciences, Changsha 410125, China. [7]Laboratory for Climate and Ocean-Atmosphere Studies, Department of Atmospheric and Oceanic Sciences, School of Physics, Peking University, Beijing 100871, China. [8]Sustainable Soils and Crops, Rothamsted Research, Harpenden AL5 2JQ, UK. [9]These authors contributed equally: Zhang Wen, Xin Ma, Wen Xu. ✉e-mail: liu310@cau.edu.cn

i.e. minimizing atmospheric pollution, yet at large economic costs. In order to sustain "blue skies", the Chinese government has issued a series of policies aimed at long-term reductions in pollutant emissions (i.e. that should persist for years), such as the release of the Air Pollution Prevention and Control Action Plan in 2013 (referred to as the Atmospheric Ten Actions), the Three-Year Action Plan for Winning the Blue-Sky Defense Battle in 2018 (referred to as the 3-year Defense Battle), and the second phase of the Action Plan. Long-term mitigation strategies are expected to be more likely to achieve environmental goals[6,7]. Large coal consumers, power plants, industry, and the transport sector were priority targets and have significantly reduced emissions of $SO_2$ and $NO_X$[8]. This has effectively mitigated acid rain and reduced $PM_{2.5}$ concentrations, as confirmed by emission inventories, environmental monitoring, and model simulations[9,10]. So is it logical and necessary to introduce short-term, provisional measures as well when long-term emission control policies appear to be working?

$PM_{2.5}$ concentrations can characterize air pollution variation at a daily scale, responding rapidly to changes in strong emissions or meteorological conditions. Atmospheric nitrogen (N) deposition, as a sink for Nr emissions, can be used as an indicator to evaluate the degree of Nr pollution at the monthly/yearly scale. Recent observations and modeling evidence show that $PM_{2.5}$ concentrations and N deposition in China experienced a rapid increase and then gradually stabilized with the implementation of long-term emission control policies[7,11–13]. In addition, global warming may contribute to changes in spatial patterns and interannual trends in atmospheric Nr emissions and N deposition due to the high correlation between temperature and $NH_3$ emissions.

Here, we aim to gain new insight into the response of air quality to Nr emission changes driven by short-term compared to long-term policies and to understand how $PM_{2.5}$ and N deposition are changing under the influence of anthropogenic activities and climate change. Our research is based on the Nationwide Nitrogen Deposition Monitoring Network (NNDMN), which contains monitoring data from nearly 60 monitoring sites with more than 40,000 samples over the 10 years from 2011 to 2020, and an integrated observation dataset of $PM_{2.5}$ and precursors collected during four international mega-events. Our analysis highlights the trade-off between short-term measures and long-term policies, and the source-sink relationships between Nr emissions and N deposition and uncovers hidden challenges in creating effective improvements in air quality.

## Results and discussion
### Short-term measures impact $PM_{2.5}$ levels
Many short-term measures have been implemented by the Chinese government to prevent air pollution from becoming a distraction during important international events, such as the suspension of manufacturing facilities and vehicle restrictions for $SO_2$, $NO_X$, and $PM_{2.5}$ emissions (Table S1). These strong, effective measures were not only implemented in Beijing but extended to surrounding regions (i.e., Tianjin and Hebei provinces). Accordingly, the concentration of major gaseous pollutants and secondary inorganic components in $PM_{2.5}$ during the Olympic Games (Olympic), Asia-Pacific Economic Cooperation Summit (APEC), and Military Parades (Parade) were significantly lower than those before the event (Fig. 1a–c). The concentrations of $SO_2$ and $NO_2$ decreased by 30%–56%, the $PM_{2.5}$ concentrations decreased by about 60%, and secondary inorganic ions (SNA: $SO_4^{2-}$, $NO_3^-$, $NH_4^+$) declined by more than 70%, indicating that the air quality was noticeably improved during the events. However, there was no substantial decrease in atmospheric $NH_3$ concentrations during each event and even a slight increase during the APEC. Also, the rigorous regulations led to a temporary pause in economic development and inconvenient living conditions for residents in the control areas. During the Belt and Road Summit (BRS), a series of "non-disturbance on residents" air quality control measures introduced by the

government produced no statistical difference in pollutant concentrations before and during the BRS, whether for $PM_{2.5}$ or precursors (Fig. 1d). Meteorological conditions need to be taken into account to determine the actual impact of the control measures on pollution mitigation, which is important for short-term case studies. Air pollutant concentrations were mostly significantly negatively linked with wind speed, precipitation, and relative humidity, but positively correlated with atmospheric pressure. The effect of temperature on air pollution has high uncertainty[14]. In this study, some of the meteorological conditions were favorable for air pollutant dispersion during the events, such as the occurrence of precipitation during the Parade and stronger southerly winds during APEC. However, emission reduction measures remained the most important driver of pollution mitigation. For example, higher temperatures, elevated relative humidity, lower atmospheric pressure, and frequent northerly winds did not reduce significant decreases in $PM_{2.5}$ concentration during the Olympic compared to the pre- and post-Olympic periods (Figs. S1 and S2).

$PM_{2.5}$ concentrations increased by 157%, 156%, and 139%, respectively, and showed no significant change (−1%) during the post-Olympic, post-APEC, post-Parade, and post-BRS periods compared to concentrations during the events. Similarly, $NO_2$ and $SO_2$ concentrations returned to "pre-event" levels, and no significant trends in $NH_3$ concentrations were observed. Except for BRS, SNA concentrations increase from "event" to "post-event", with a 5-fold growth after the Parade. Similarly, a decrease of 12% in $PM_{2.5}$ and 12% in SNA concentrations was observed for the Hangzhou G20 Summit compared to the period before aggressive mitigation measures were introduced. However, after the various control measures were lifted, all air pollutant concentrations increased back to initial levels[15]. During the 2022 Beijing Winter Olympics, the intense emission controls significantly reduced $PM_{2.5}$ and SNA concentrations, with source control measures contributing 54% to the total reduction in $PM_{2.5}$ concentrations[16]. In summary, "Olympic Blue", "APEC Blue" and "Military Parade Blue" impressed the world, and the "blue skies" increased people's determination to defeat air pollution and showed the potential for air quality improvement. Mandatory administrative measures to reduce air pollution, such as those in Table S1, were temporary and unsustainable in the long term. However, such short-term abatement measures are essential because the moral incentive probably outweighs the rebound after the decrease in $PM_{2.5}$ concentration during the event. This means that, in order to improve air quality standards in the short term, it is necessary to adopt some unconventional emission reduction measures at the expense of the economy to remind the public that pollution can be prevented and so increase their confidence in policies and practices.

From August 2005 to August 2020, the concentrations of $PM_{2.5}$, SNA, and gaseous precursors in Beijing decreased by 68%, 66%, and 52%, respectively (Fig. 1e). Notably, the reduction in $SO_2$ was as high as 90%, while the decreases in $NO_2$ and $NH_3$ were limited to 38% and 26%. Although $PM_{2.5}$ concentrations in Beijing are significantly higher in winter than in summer, they still showed a continuous downward trend over the 15 years: from January 2005 to January 2020, $PM_{2.5}$, SNA, $SO_2$, and $NO_2$ decreased by 65%, 51%, 94%, and 50%, respectively (Fig. 1f). In contrast to the summer months, $NH_3$ concentrations increased by about 50% in winter and stabilized at $7\,\mu g\,m^{-3}$ after 2013. China effectively controls $SO_2$ emissions through strengthened industrial emission standards, upgrades on industrial boilers, and the phasing out of outdated industrial capacity[7]. The "Olympic Blue" in August 2008 was a phenomenon that astonished the nation at the time. In August 2009, a significant rebound in the concentrations of $PM_{2.5}$, ionic components, and gaseous precursors was observed, returning to the levels documented in August 2007, which underscored the limitations of the short-term emission reduction measures. Nonetheless, after approximately six years, the concentrations of

 

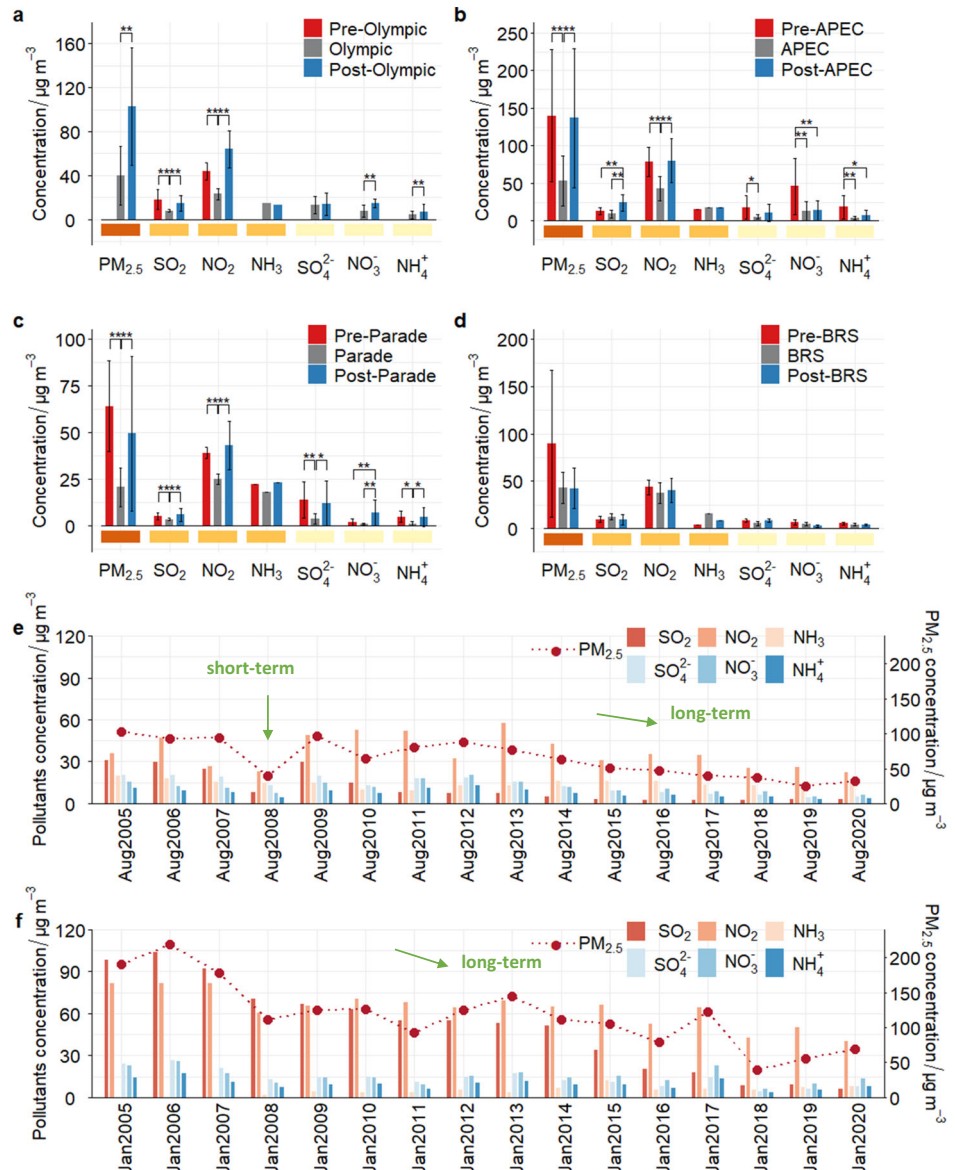

**Fig. 1 | PM$_{2.5}$, gaseous precursors (SO$_2$, NO$_2$, NH$_3$) and secondary inorganic ion (SNA: SO$_4^{2-}$, NO$_3^-$, NH$_4^+$) concentrations (average ± SD) in Beijing. a** pre-Olympic Games (Olympic) (Jun. 1–30, 2008), Olympic (Aug. 8–24, 2008) and post-Olympic (Sep. 24–Oct. 23, 2008); **b** pre-Asia-Pacific Economic Cooperation Summit (APEC) (Oct. 1–31, 2014), APEC (Nov. 1–12, 2014) and post-APEC (Nov. 13–Dec. 31, 2014); **c** pre-Military Parades (Parade) (Aug. 3–19, 2015), Parade (Aug. 20–Sep. 3, 2015) and post-Parade (Sep. 3–29, 2015); **d** pre-Belt and Road Summit (BRS) (Apr. 28–May 7, 2017), BRS (May 8–17, 2017) and post- BRS (May 18–Jun. 17, 2017 May 18–Jun. 17, 2017); dark orange in X axis - PM$_{2.5}$ concentration, orange - gas precursor concentrations, light orange - ionic concentrations in PM$_{2.5}$; **e** the period of August 2005 to August 2020; **f** the period of January 2005 to January 2020. Source data are provided as a Source Data file.

pollutants in all subsequent years were consistently lower than those recorded in 2008. Similarly, the air pollution episode in January 2013 caused strong dissatisfaction among the government and the public of Beijing. However, from that time until 2020, the PM$_{2.5}$ concentration in Beijing decreased by approximately 54%, demonstrating the effectiveness of sustained emission control policies, including the Atmospheric Ten Actions (2013–2017) and the 3-year Defense Battle (2018–2020).

Short-term measures were virtually unaffected by long-term emission reductions, with short-term net effects of 67% and 61% of APEC and Parade events, respectively. Short-term measures also did not inherently alter the slope of the decline in long-term pollution reduction ($p > 0.05$, Note S1). This is because the decrease in concentrations from long-term policies is stable and gradual. The >50% reduction in PM$_{2.5}$ concentration achieved in Beijing through one

month of short-term measures has in fact taken nearly a decade to stabilize. With continuous adjustments in energy and industrial structures, the concentration of atmospheric pollutants is expected to continue to decline. When the concentration of pollutants is extremely low, it shows that long-term emission reduction policies have played an extremely significant role, and so the implementation of short-term measures might be ineffective and unnecessary under this condition.

## Long-term controls affect N deposition trends

The average concentration of atmospheric Nr (NH$_3$, NO$_2$, HNO$_3$, $p$NH$_4^+$, $p$NO$_3^-$) over the 10 years from 2011 to 2020 was 22.3 ± 1.34 µg N m$^{-3}$. It peaked in 2013, and then slowly decreased by 19% to 2020 (Fig. 2a). The average Nr concentration in precipitation was 2.86 ± 0.39 mg N L$^{-1}$ (NO$_3^-$, NH$_4^+$), with the highest value occurring in 2012, decreasing by 36% to 2020 (Fig. 2b). The site-average dry, bulk and total N deposition

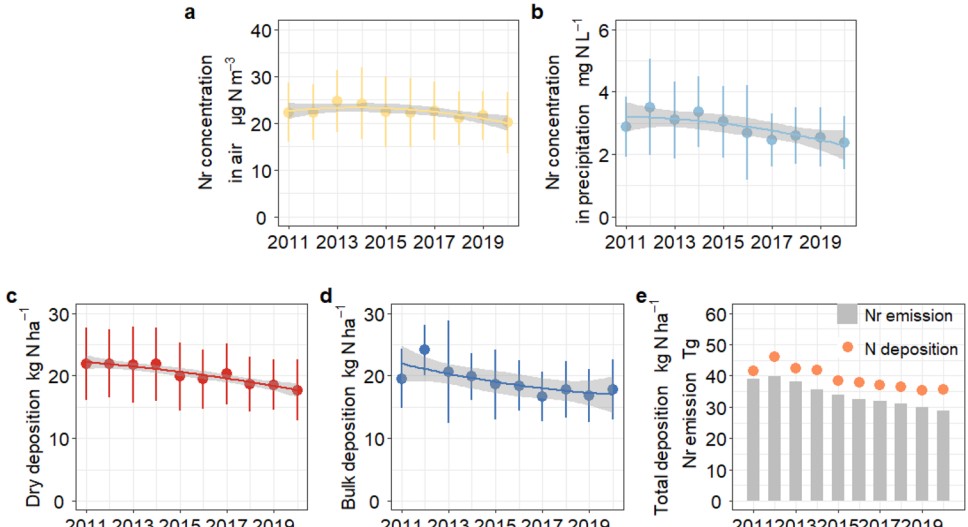

**Fig. 2 | Interannual variations of Nr concentrations and N deposition (average ± SD) in China. a** Nr concentration in air, **b** Nr concentration in precipitation, **c** dry deposition, **d** bulk deposition, **e** total deposition, and Nr emission. Source data are provided as a Source Data file.

in China were estimated at $20.2 \pm 1.60$, $19.0 \pm 0.82$ and $39.2 \pm 2.42$ kg N ha$^{-1}$ yr$^{-1}$ during the 10 years, respectively. Given the spatial heterogeneity of monitoring sites, the geographic annual mean fluxes were $14.8 \pm 1.71$, $12.8 \pm 0.82$ and $27.7 \pm 2.42$ kg N ha$^{-1}$ yr$^{-1}$ for dry, bulk, and total N deposition, which are in good agreement with satellite observations and model simulations[17,18]. Annual N deposition was estimated to continuously decline in China by about 19%, 26%, and 23% for dry, bulk, and total deposition from 2012 to 2020, respectively (Fig. 2c–e, $p < 0.01$). A strong positive relationship was observed between Nr emissions and N deposition, implying a rapid response of N deposition to changes in emissions ($R^2 = 0.90$, Fig. 2e). These trends are consistent with the strict control of Nr emissions since 2012, which caused annual NO$_X$ emissions to decline by 33% from 2012 to 2020 (MEIC, http://www.meicmodel.org). However, NH$_3$ emissions declined only from 10.6 to 9.07 Tg during the same period due to the few national policies implemented for NH$_3$ emissions control. In contrast to PM$_{2.5}$ concentrations, monthly averaged atmospheric N deposition was significantly higher in summer than in winter by 32% (Fig. S3). However, similar to PM$_{2.5}$ concentrations, under the influence of long-term emission reductions, the alleviation of summer pollution is greater than that in winter. Winter air quality clearly needs to be improved.

According to a Mann–Kendall test of the data, the decline in N deposition slowed in 2015. We divided the 10 years into 3 stages, 2011–2014 (period of severe air pollution), 2015–2017 (period of air pollution mitigation), and 2018–2020 (period of significant air quality improvement). The changes in dry, bulk, and total N deposition from 2011–2014 to 2015–2017 were −14%, −13%, and −13% (consistent with the reductions in Nr emissions of 14% during the same period), followed by smaller reductions from 2015–2017 to 2018–2020 of −11%, 1.3%, and −5.2% (Fig. S4). Similar trends are revealed in the interannual variation of Nr emission inventories, implying a decline in emission reductions in recent years, as the most easily achievable reductions, especially for NO$_X$, had already worked through in the early stages of pollution emission control[19,20]. Future air quality improvement needs an even greater focus on technological innovation to create breakthroughs, which has not been emphasized in previous research. Compared to immediate, effective but short-lived short-term pollution control measures, long-term policies take more time to have an impact, but their effects are more permanent. Even though the decline in N deposition and the improvement in air quality slowed over the 10 years, there was no significant rebound.

The components and forms of Nr concentrations and N deposition have undergone significant changes over the 10-year study period, revealing new challenges for future air pollution control. NH$_3$ was the largest component of total deposition (26%) during 2011-2020, followed by bulk NH$_4^+$, bulk NO$_3^-$, HNO$_3$, NO$_2$ and $p$NH$_4^+$, and $p$NO$_3^-$ (Fig. 3a). The reduced N (NH$_x$, NH$_3$, $p$NH$_4^+$) in dry deposition plus NH$_4^+$ in bulk deposition were therefore the main forms deposited (57%) in total deposition. As shown in Fig. 3b, the dry deposition of NO$_2$, HNO$_3$, $p$NH$_4^+$, and $p$NO$_3^-$ significantly decreased (−36%, −64%, −33%, −47%) from 2011 to 2020, whereas that of NH$_3$ increased. The decline in oxidized N deposition (NO$_y$: NO$_2$, HNO$_3$, $p$NO$_3^-$ in dry deposition plus NO$_3^-$ in bulk deposition) is in agreement with the decline in NO$_X$ emissions (of NO$_y$ deposition by 31% vs NO$_X$ emission by 33%, Fig. 3c). The proportion of HNO$_3$ and NO$_2$ gaseous forms in total oxidized N deposition decreased from 0.45 in 2011 to 0.28 in 2020 ($p < 0.01$), but the variation in dry oxidized deposition was negligible (Fig. S5). The percentage of HNO$_3$ decreased while that of NO$_2$ increased annually, which may be related to the homogeneous and heterogeneous photolysis of nitrate in the atmosphere[21–23]. In contrast, the gaseous fraction of NH$_3$ increased from 0.39 to 0.52 over the 10 years. Ambient mean NH$_3$ concentrations and NH$_3$ deposition in 2020 were 34% and 27% higher than in 2011, counteracting the decrease in $p$NH$_4^+$ concentrations, confirming the stability of NH$_3$ emissions and indicating that more NH$_3$ is present as a gas in the atmosphere due to the reductions in acid gas emissions[24,25]. The decrease in NH$_4^+$ bulk deposition (from 10.3 to 7.80 kg N ha$^{-1}$ yr$^{-1}$, $p = 0.01$) offset the increase in gaseous NH$_3$ so that total reduced N deposition presented a small decrease, which was generally consistent with the change in NH$_3$ emissions (NH$_x$ deposition decreased by 13% vs NH$_3$ emission by 14%, Fig. 3d). The NH$_3$-rich atmospheric environment ($AdjGR > 1$, defined in Note S2) and the small increase of NO$_3^-$ bulk deposition (8%, $p > 0.05$) show that NO$_X$ emission reduction policies need to be more stringent and placed alongside NH$_3$ emission controls, especially for non-point source emissions, as opportunities for acid gas emission reduction decrease.

The 10-year average N deposition at urban, rural, and background monitoring sites was $47.8 \pm 7.32$, $39.3 \pm 2.32$, and $24.6 \pm 6.49$ kg N ha$^{-1}$ yr$^{-1}$, respectively. Deposition of NH$_x$ and NO$_y$ at urban monitoring sites was 10% and 37% higher than at rural sites. Annual N deposition exhibited different interannual variability at the three land-use types (Figs. 3e–g and S6). NH$_x$ and NO$_y$ deposition at urban monitoring sites decreased at an annual rate of 3.5% and 3.9%

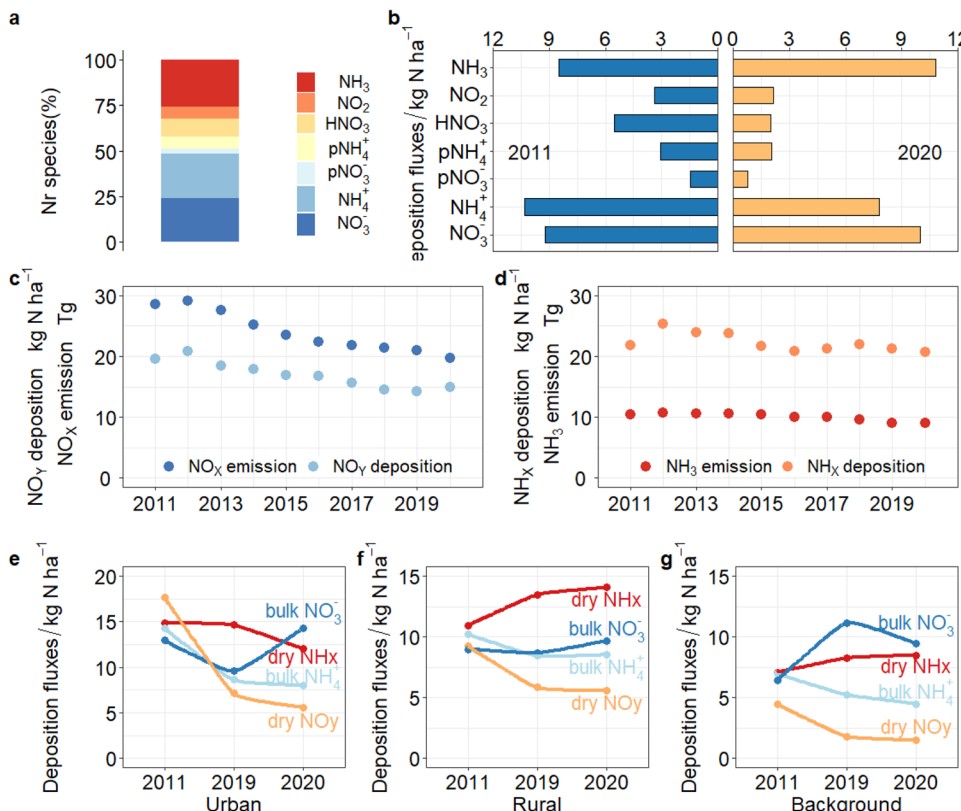

**Fig. 3 | Interannual variations of reduced and oxidized N deposition.**
**a** Percentage of each species in Nr deposition; **b** Nr species deposition in 2011 and 2020; **c, d** the relationship between $NO_x$ emissions and NOy deposition, $NH_3$ emission and $NH_x$ deposition; **e**–**g** deposition at urban, rural, and background areas. Source data are provided as a Source Data file.

from 2011 to 2020 ($p < 0.01$). At rural monitoring sites, $NO_y$ deposition declined more slowly because the energy structure transformation and stricter $SO_2$ and $NO_x$ emission standards of industry and transportation mainly affected urban areas. $NH_x$ deposition showed irregular variation in rural areas because the slight increase in $NH_3$ dry deposition offset the decrease in $NH_4^+$ bulk deposition. Air pollutant emissions were greatly reduced at national scales during the coronavirus 2019 pandemic lockdown period (COVID-19). The mean dry deposition of reduced N continued to increase from 2019 to 2020 in rural regions but decreased at urban sites. One possible explanation is that non-agricultural emissions were the major $NH_3$ sources in urban areas but agricultural $NH_3$ volatilization resulted in persistent high values in rural areas[26,27].

Agricultural sources account for more than 80% of total $NH_3$ emissions, so the control of $NH_3$ emissions from non-point sources is imperative[28,29]. According to Gu et al.[30], the cost of technological measures to reduce global agricultural $NH_3$ emissions by 32% is around 19 billion USD. However, the technologies mentioned do not take into account the difficulty of technology diffusion and acceptability for farmers, so the actual cost of reducing $NH_3$ emissions from agricultural sources may be higher than this. Very few studies have addressed urban $NH_3$ emission reduction actions. Nonetheless, with adjustments in urban industrial structures and the promotion of electric vehicles and other measures that are mainly focused on $NO_x$ and other pollutants, it is possible to achieve a "free ride" on reducing urban $NH_3$ emissions, which could significantly lower the economic cost of urban $NH_3$ emission reduction.

### Regional differences and transboundary effects
Driven by the effects of short-term policy implementation and an earlier and broader transformation of energy structures in North China

(NC), this region saw a more dramatic abatement of Nr emissions than other regions, a faster annual reduction in N deposition at −3.9% since 2011 ($p < 0.01$), and a clear decoupling of N deposition and economic development (Fig. 4a). In contrast, the changes in total N deposition in other regions were not significant (Fig. 4b). The decrease in the dry deposition of oxidized N ranged from 43 to 57% across all regions, except for Northeast China (NE) where it was reduced by about 21%. For $NH_x$ deposition, the spatial analysis showed more complex features. The concentration of $NH_3 + pNH_4^+$ in NC increased until 2014 and leveled off after 2015, and in Southwest China (SW), $NH_3 + pNH_4^+$ showed non-significant changes until 2017 and then increased. The trends in oxidized N concentrations can explain the differences in $NH_3 + pNH_4^+$ concentrations between the two regions. Oxidized N concentrations have continued to decline in NC (both in air and precipitation). This suggests that $NH_3$ emissions have declined significantly in this region in recent years, causing the smaller oxidized N concentrations and the stabilization of $NH_3 + pNH_4^+$ concentrations. In contrast, the $NH_3$ emission inventory for the Southwest region shows a 10% decrease from 2017–2020. Therefore, an observed increase in atmospheric $NH_3 + pNH_4^+$ concentrations may be attributed to a decrease in acid gas emissions ($SO_2$ emission declined by 36%, $NO_x$ emission decline by 11%) and fewer neutralization reactions, resulting in more $NH_3$ present in the atmosphere. Inter-regional differences in $NH_3$ emissions are understandable. The decreased $NH_3$ emissions in NC are due to the previous agricultural policies introduced by the government, e.g., the Three-year action plan to win the Blue-Sky Protection Campaign (https://www.gov.cn/zhengce/content/2018-07/03/content_5303158.htm), and the promotion of numerous $NH_3$ emission reduction measures, e.g., the reduction of farmland $NH_3$ volatilization through better controls on N-fertilizer applications, and the introduction and promotion of urease inhibitors[31,32].

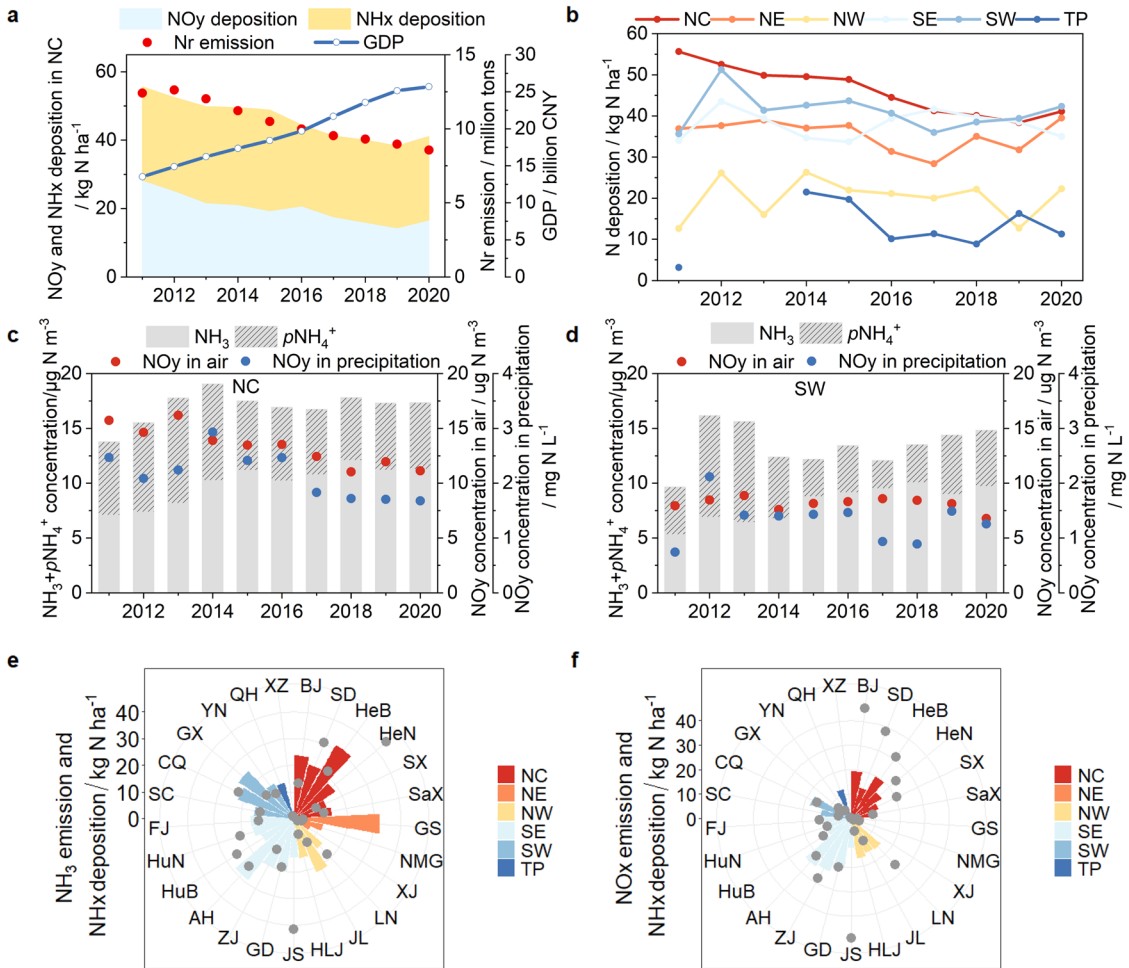

**Fig. 4 | Characteristics of interannual variability of N deposition in different regions. a** N deposition, Nr emission, and Gross Domestic Product (GDP) in North China (NC); **b** interannual variation of N deposition in NC, Northeast China (NE), Northwest China (NW), Southeast China (SE), Southwest China (SW), and the Tibetan Plateau (TP); **c**, **d** NH₃ and $p$NH₄⁺ concentrations in the air, oxidized N concentrations in the air and precipitation over NC and SW; **e**, **f** differences in N deposition and Nr emissions at the provincial scale (BJ Beijing, SD Shandong, HeB Hebei, HeN Henan, SX Shanxi, SaX Shaanxi, GS Gansu, NMG Inner Mongolia, XJ Xinjiang, LN Liaoning, JL Jilin, HLJ Heilongjiang, JS Jiangsu, GD Guangdong, ZJ Zhejiang, AH Anhui, HuB Hubei, HuN Hunan, FJ Fujian, SC Sichuan, CQ Chongqing, GX Guangxi, YN Yunnan, QH Qinghai, XZ Tibet). Source data are provided as a Source Data file.

The spatial correlations of N deposition and Nr emissions vary from one region to another (Fig. 4e, f). NH₃ emissions and NHₓ deposition were higher in NC, Southeast and SW, consistent with regions that have frequent and intensive agricultural activities. NH₃ emissions and NHₓ deposition were generally comparable within regions, exhibiting near-source deposition of NH₃. By comparison, the highest NO$_y$ deposition is mainly concentrated in northern China, such as Beijing, Shandong and Hebei provinces, with their highly developed industry and transportation sectors. Topography and meteorological conditions determine that the spatial characteristics of the different air pollutants in China are very similar, and N deposition is no exception. For NC, NO$_X$ emissions were generally higher than NO$_y$ deposition at the provincial scale, while the surrounding areas show the opposite relationship. The gap between them implies that there is trans-boundary export of pollution from the NC: about 50% of NO$_X$ is deposited in the region and the rest is "exported" to the northwest or southeast regions. Pollution transport not only affects N inputs to ecosystems in the surrounding area, but also harms human health. A recent study showed that approximately half of the PM₂.₅-related mortality rates were attributed to emissions from sources outside the regions in which the deaths occurred[33,34]. This shows the importance of strengthening the joint prevention and control of air pollution at the regional level.

## Future air quality improvement potential

It is now accepted that global temperatures will increase slowly and extreme precipitation will intensify in the future[35] (https://www.ipcc.ch/sr15/), and that improvements in air quality could be negated by climate change[36]. In this study, we found the effect of meteorology is more clearly reflected in the monthly changes of N deposition (Fig. S7), with precipitation and temperature being the significant explanatory variables (Fig. S8). Wang et al.[37] discovered a previously unrecognized mechanism showing that synoptic climate change might have a bigger effect in increasing the concentration of PM₂.₅ in low quartiles than in upper quartiles. Meanwhile, the Chinese government will not relent in its efforts to address air pollution issues, with its commitment to "Carbon Emission Peaking in 2030" and "Carbon Neutrality in 2060".

We simulated the response of air quality to changes in precursor emissions: a base simulation ("Base: 2017") and two scenarios ("Case 1: 2030", Carbon Emission Peaking; "Case 2: 2060, Carbon Neutrality") were developed to quantify the contribution of near-term (7–10 years) or long-term (30–40 years) emission reductions to PM₂.₅ concentrations and N deposition (simulation details are in the "Methods" section and Table S2). Atmospheric emissions of acid gases and NH₃ are expected to decline by, respectively, ~40% and ~10% by 2030, and ~90% and ~50% by 2060, with the increasing impacts of low-carbon policies (Fig. S9). Overall, the control of agricultural emissions is lagging

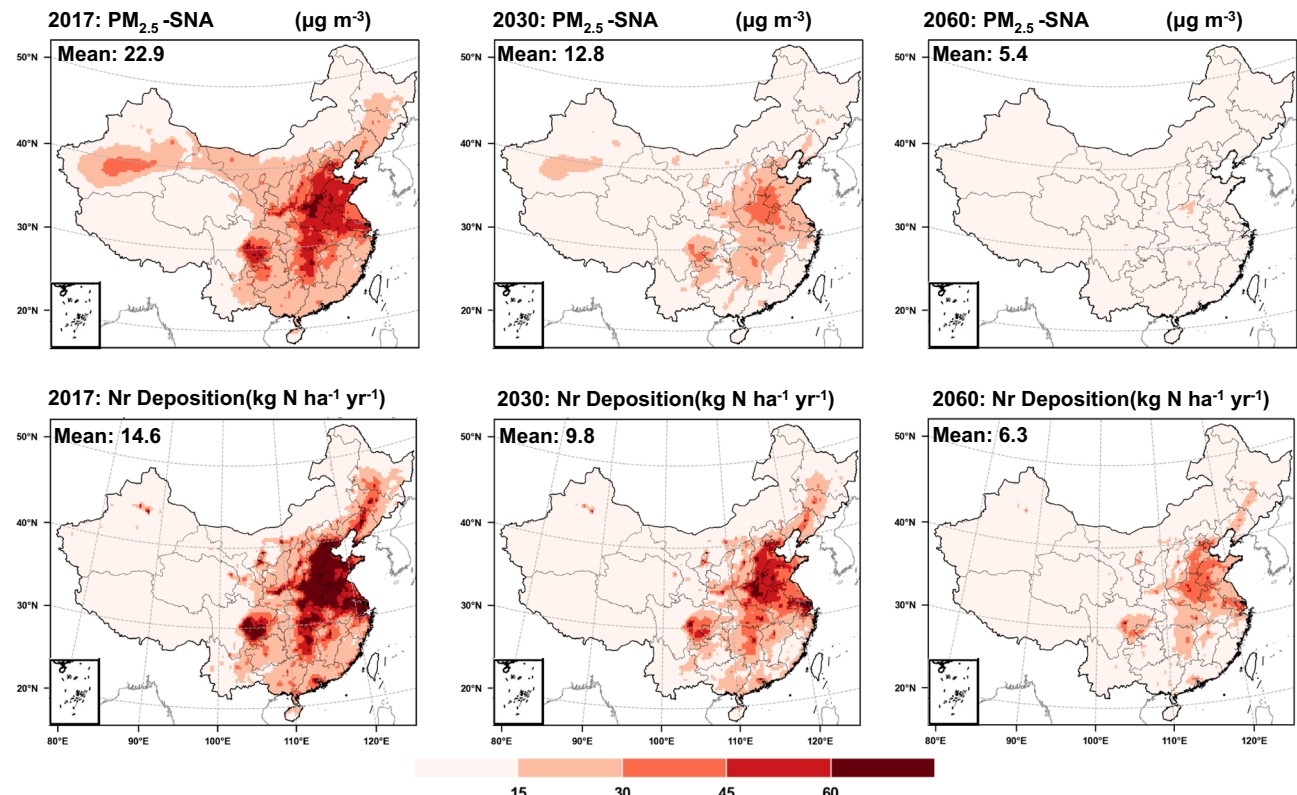

**Fig. 5 | Annual average PM_{2.5}-SNA concentrations and N deposition in China in 2017, 2030, and 2060, using the WRF-EMEP model under the most stringent policy implementations and rising temperatures and increased extreme precipitation (RCP4.5_2030, RCP4.5_2060) in 2030 and 2060.** The map data are provided by the Resource and Environment Science and Data Center Institute of Geographic Sciences and Natural Resources Research, Chinese Academy of Sciences (https://doi.org/10.12078/2023010103, 2023).

behind the implementation of stringent regulations on power plants, industry and vehicle emissions. PM_{2.5}-SNA concentrations are predicted to decrease significantly by 2030 and 2060, and the national average concentrations to decrease by 44% and 76% compared to 2017, respectively (Fig. 5). The national percentage area exceeding 15 µg m$^{-3}$ PM_{2.5}-SNA is predicted to be 51%, 20% and 0.3% in 2017, 2030 and 2060, respectively. In contrast, the response of N deposition to emission reductions is weaker, with the mean N deposition decreasing by 33% and 57% by 2030 and 2060, respectively, compared to 2017, which illustrates the complex multifactorial processes occurring from precursor release to gas-particle transformation, and deposition.

In view of the impact of climate change on air quality (global warming and increased extreme precipitation), policies and the feasibility of emission reduction technologies need to be rationalized. We have shown that emission control policies are effective, both in the short term for preventing haze pollution during important international events and in the long term for national air quality improvement. Even though there is a significant rebound in air quality after short-term, campaign-style political controls, the effectiveness of such approaches has greatly increased public confidence in obtaining "daily blue skies", i.e. cleaner air, and has shown the potential for air quality improvement. This is particularly important for heavily polluting developing countries[38]. Compared to short-term policies that sacrifice economic benefits and cause some public inconvenience, long-term policies are more likely to achieve the dual objectives of not damaging economic development and improving environmental protection[39]. However, they inevitably take several years to have an impact, with the risk that those involved in implementation lose confidence and patience. Therefore, a combination of long-term and short-term emission controls is most appropriate for air quality improvement, delivering an immediate and observable effect of short-term measures

that reduce pollutant emissions, and the stabilizing effect of long-term policies contributing to the sustainability of cleaner air.

## Policy implications

In contrast to acid gas emission reductions, which can be achieved with both short-term measures and long-term policies, the control of non-point source NH_3 emissions can only be implemented through agricultural innovations that improve crop N use efficiency and reduce environmental impacts. Currently, the measures for air quality control mainly focus on end-of-pipe treatments and cleaner production, which are beneficial for air pollution emission control targets but may not be able to reach the dual objectives of air quality improvement and climate change mitigation[40,41]. Therefore, policymakers should promote sustainable approaches impacting all necessary sectors and objectives, such as low/zero/negative-carbon production methods for industry through raw material, product structure and production equipment upgrades, ultra-low pollutant emissions from transportation through the popularization of electric vehicles and synergistic pollution control on diesel trucks, aircraft and ships, and green development in agriculture through the application of precise management in crop breeding and production[42–44].

Compared to current policies, three trade-off relationships should be considered for policy development. One is the already-mentioned trade-off between inspiring but aggressive short-term measures and effective but durable long-term policies. The combination of long-term and short-term emission control policies is the sustainable solution for air quality improvement. Frequent or prolonged implementation of high-intensity short-term control measures will disrupt the development of industry, transport and residential life. Therefore, at a time when the effectiveness of long-term emission control is still tentative, a short but major emission control exercise should be carried out once

every 1–2 years to clarify the regional emission reduction potential. Policymakers should gradually adjust the intensity of the controls, and align them with long-term goals, ensuring a smooth transition and continuous progress towards objectives, defining specific and achievable long-term targets for air quality improvements that are informed by the success of short-term reductions. In the long term, two aspects need to be clarified. Firstly, "polluter identification". For example, sintering in the steel industry is a major source of $NO_X$ emissions, while in agriculture, high amounts of $NH_3$ are emitted from the treatment of manure from livestock. Therefore, to increase pollution management efficiency, the specific targets of emission reduction must be identified (including key emitting sectors and critical emission stages). Secondly, the principle of "polluter pays" should be achieved. To ensure fairness in environmental and economic terms within pollution management, it is essential to identify the input and output of pollutants between regions. There needs to be a more precise quantification of transboundary contributions, which can provide data support for subsequent compensatory mechanisms between regions.

The second is the oxidized – reduced N deposition/dry – wet N deposition trade-off, as the reduction in oxidized N dry deposition is offset by the increase in bulk deposition, and the decrease in acid gas emissions increases $NH_3$ concentrations. To avoid this, synergistic controls of multiple precursors are necessary. Compared to $SO_2$ and $NO_2$, it is almost impossible to achieve rapid reductions in $NH_3$ concentrations quickly. Our case study during the COVID-lockdown period has proven that, after a drastic reduction of transport and industrial $NH_3$ emissions in February 2020, ambient $NH_3$ concentrations from January to February (17%) still exceeded the levels of previous years (9%), while concentrations of $SO_2$ and $NO_2$ decreased by 24% and 49%, respectively[26]. This indicates that, as opposed to the intense reductions of acid gases, $NH_3$ emissions can only be gradually reduced through better fertilizer and manure management. Due to the lack of a national $NH_3$ monitoring system overseen by the government, the impact of $NH_3$ on air quality and the environment remains a matter of theoretical debate[45]. Currently, "Green Demonstration Zones" have been established across China, where systemic transformations in precision and smart agriculture will improve nitrogen use efficiency and reduce $NH_3$ volatilization. We suggest that research on $NH_3$, $PM_{2.5}$ and atmospheric N deposition should be uniformly strengthened in this area, treating spatial "regional areas" analogously to temporal "short-term" studies, thereby identifying the national potential and environmental effects of $NH_3$ emission reduction through case studies.

Third is the trade-off between inter-regional Nr pollution and controls. Regional-specific and cross-regional prevention measures are needed in the face of spatial heterogeneity and transboundary air pollution. In the case of cross-boundary pollution transport, the implementation of stricter emission standards and production technology innovation could deliver win-win air pollution reductions for origin and destination regions[46]. Our regionalization results will help in the design of region-specific pollution control strategies that should be further investigated. Moreover, it is important to include public opinion in the process of strategy- and policy-making and implementation, balancing the desire of citizens to improve air quality against ensuring the normal conduct of their lives[47]. In particular, for the control of non-point source $NH_3$ emissions, and to achieve comprehensive and sustainable air quality improvement, it is crucial to focus on targeted strategies that promote sustainable agricultural practices and involve collaboration between all stakeholders, including government agencies, farmers and the public. Quantifying the economic benefits and environmental costs associated with pollution transport contributes to revealing the environmental inequalities of regional emissions and pollution. As reflected in this study, economically developed regions export Nr emissions to neighboring regions,

which results in these suffering both economic and environmental losses. Currently, economic compensation between regions under multiple constraints (air pollution, climate change, ecological effects) is hardly discussed. Regions more economically developed are encouraged to increase green investments in emission-reducing technological transformations[48] and may need to compensate polluted regions around them. This could be through economic measures such as tax incentives, subsidies and loans, encouraging enterprises and consumers to make more environmentally friendly choices. For instance, financial support could be provided to enterprises and farms that utilize clean energy and energy-efficient technologies, and tax relief given to consumers for purchasing environmentally friendly products and services. Our work also shows that long-term monitoring is essential for an effective understanding of environmental change and air quality improvement that is not possible with short-term studies alone.

## Methods

### Daily average $PM_{2.5}$ and its precursor concentrations

Daily average $PM_{2.5}$ concentrations and ionic components during the Olympic Games, APEC and Military Parades periods were obtained from published literature[49–51]. $PM_{2.5}$ concentrations and ionic concentrations during the Belt and Road Summit were taken from our group observation results. Daily concentrations of gas precursors ($SO_2$, $NO_2$) were obtained from the China National Environmental Monitoring Centre (CNEMC, http://www.cnemc.cn/sssj/). $PM_{2.5}$ and ionic component concentrations are from Tracking Air Pollution in China (TAP, http://tapdata.org.cn/). Since $NH_3$ data are unavailable from CNEMC, they were taken from our group's monthly sampling results and used as an approximate substitute for atmospheric $NH_3$ concentrations during events. ALPHA passive samplers were used to obtain monthly $NH_3$ samples for the four study periods (Adapted Low-cost High Absorption, Center for Ecology and Hydrology, Edinburgh, UK). Detailed measurement and analysis procedures for the ALPHA samplers are described in Note S3. The durations of the pre-event, event window and post-event for each event are listed in Table S3.

### Monthly Nr deposition estimations

The monitoring of pollutants in Beijing began with the measurement of acid rain in 1998. The Nationwide Nitrogen Deposition Monitoring Network (NNDMN) has been in operation since 2011 and collects samples, including gaseous and particulate Nr and precipitation. Ambient gaseous ($NH_3$ and $HNO_3$) and particulate ($pNH_4^+$ and $pNO_3^-$) Nr samples were collected monthly with active DELTA samplers (Denuder for Long-Term Atmospheric sampling, Center for Ecology and Hydrology, Edinburgh, UK), and $NO_2$ samples were collected monthly using Gradko passive diffusion tubes (Gradko International Limited, UK). $NH_3$ was also sampled using an ALPHA passive sampler at all monitoring sites as a check on results. Passive samplers were deployed in duplicate or triplicate. Precipitation samples were collected after each precipitation event in rain gauges (SDM6, Tianjin Weather Equipment Inc., China).

Gaseous and particulate samples were stored at 4 °C, and precipitation samples stored at −4 °C until analyzed. More details of the sample processing and quality control procedures are shown in Note S3. Dry N deposition ($F_d$) was calculated as the product of the Nr concentration ($C_d$) and deposition velocity ($V_d$) (Eq. (1)). Hourly $V_d$ for 5 Nr species were simulated by the GEOS-Chem chemical transport model (Note S4, Fig. S10)[52]. Innovations in industrial or agricultural technology could directly alter the emissions and concentrations of Nr, while the stability of land-use types maintains the resistance of gases to capture by surface plants or soil and so to $V_d$, thereby emphasizing the key impact of Nr concentration on the trend of dry deposition. Bulk N deposition ($F_b$) was calculated based on the Nr

concentration ($C_b$) and precipitation amount ($P_t$) (Eq. (2)).

$$F_d = C_d \times V_d \tag{1}$$

$$F_b = P_t C_b / 100 \tag{2}$$

$$F = F_d + F_b \tag{3}$$

The 59 sites were selected from the NNDMN for data analysis (sampling period of more than one year) to cover major land-use types: urban, cropland, coastal, forest and grassland. More details on each site including location, monitoring period, monitoring method and land-use type can be found in Table S4. We analyzed the spatial patterns of dry/bulk deposition using both arithmetic averaging of the sites and weighting based on regional area to avoid overestimation or underestimation of deposition fluxes due to uneven size distribution (Eq. (4), Eq. (5)).

$$\begin{aligned}
\textit{Site average N deposition} = (&F_{NC1} + F_{NC2} + \cdots + F_{SE1} + F_{SE2} + \cdots + F_{SW1} \\
&+ F_{SW2} + \cdots + F_{NW1} + F_{NW2} + \cdots + F_{NE1} \\
&+ F_{NE2} + \cdots + F_{TP1} + F_{TP2} + \cdots) / 59
\end{aligned} \tag{4}$$

$$\begin{aligned}
\textit{Geographic average N deposition} = &\left[ (F_{NC1} + F_{NC2} + \cdots) / \left( \frac{A_{NC}}{960} \right) \right] \\
&+ \left[ (F_{SE1} + F_{SE2} + \cdots) / \left( \frac{A_{SE}}{960} \right) \right] \\
&+ \left[ (F_{SW1} + F_{SW2} + \cdots) / \left( \frac{A_{SW}}{960} \right) \right] \\
&+ \left[ (F_{NW1} + F_{NW2} + \cdots) / \left( \frac{A_{NW}}{960} \right) \right] \\
&+ \left[ (F_{NE1} + F_{NE2} + \cdots) / \left( \frac{A_{NE}}{960} \right) \right] \\
&+ \left[ (F_{TP1} + F_{TP2} + \cdots) / \left( \frac{A_{TP}}{960} \right) \right]
\end{aligned} \tag{5}$$

$F_{NCi}$ represents total N deposition at the NCi site, and $A_i$ represents area of region i.

For the quantification of pollutant transport between regions, we assumed that Nr emissions and deposition in China are balanced, with no pollutants being transported from China from other countries; the emission-to-deposition between regions is defined simply as all pollutants will preferentially be deposited in their own region.

### Factors vary N deposition
Random forest (RF) evaluates the complex relationships between the response variable and interpretation variables and has been widely used for the prediction of air pollution or identifying key determining factors. The random forest modeling program was run by using the "randomforest" package in R software (version 3.6.1). We constructed the functions of "oxidized /reduced /total N deposition ~ acid gas emissions + ammonia emission + meteorological parameters + soil parameters + NDVI". The robustness of the RF model was tested by 10-fold cross-validation.

Monthly oxidized, reduced and total N deposition was derived from the NNDMN. Monthly $SO_2$, $NO_X$ and $NH_3$ emissions from 2011 to 2020 in China were obtained from the Model of Multi-resolution Emission Inventory (MEIC, http://www.meicmodel.org). Meteorological parameters included precipitation amount, temperature at 2 m, wind speed at 10 m, surface pressure and boundary layer height. Soil parameters included soil temperature, soil moisture and evapotranspiration. The monthly precipitation amount was obtained from NNDMN, and other meteorological and soil factors were downloaded from the European Centre for Medium-Range Weather Forecasts (https://apps.ecmwf.int, 0.25° × 0.25°). Data for NDVI were derived from a Moderate Resolution Imaging Spectroradiometer (MODIS) Level-3 product (https://neo.sci.gsfc.nasa.gov, 0.1° × 0.1°). The deposition at each sampling site corresponds to emissions, meteorological, soil and NDVI parameters through latitude and longitude.

### The future trends in $PM_{2.5}$ concentrations and N deposition
The nonlinear responses of $PM_{2.5}$ concentrations and N deposition to the precursor emissions ($SO_2$, $NO_X$, $NH_3$, etc) and meteorological factors were simulated using the WRF (version 3.9.1) - EMEP (version rv4.17) model system (27 km × 27 km). The meteorological conditions were obtained from Final Operational Global Reanalysis (FNL) data and the Representative Concentration Pathway (RCP). For anthropogenic emissions, the MEIC and Asian anthropogenic emission inventory (MIX) were used. Here, the ground observation data of $PM_{2.5}$ concentrations and N deposition were used to evaluate the model performance. $PM_{2.5}$ concentrations were taken from TAP, and the spatial correlation $R^2$ between them reached 0.77 in 2017 (Fig. S11). The N deposition simulation results were calibrated against NNDMN monitoring data from 2008 to 2017[18].

To evaluate the impact of anthropogenic emissions and climate change on $PM_{2.5}$ concentrations and N deposition, we performed the following simulation scenarios using the WRF-EMEP model: Base scenario: the scenario to simulate the spatiotemporal of $PM_{2.5}$ concentrations and N deposition in 2017; the meteorological conditions and anthropogenic emissions were kept at the level of 2017. Case 1 (2030, Carbon Emission Peaking) / Case 2 (2060, Carbon Neutrality): these scenarios simulated the spatiotemporal $PM_{2.5}$ concentrations and N deposition under the most stringent policy implementations at these two future dates, with climate change taken into account, with rising temperatures and increased extreme precipitation (RCP4.5_2030, RCP4.5_2060).

### Data analysis
A one-way analysis of variance (ANOVA) and nonparametric $t$-tests were used to examine the differences between land-use types and regions. A Mann–Kendall (MK) test was conducted to analyze the trends in N deposition and Nr emission and Sen's slope was used to calculate the trend slope. Linear or nonlinear regression analyses related to bulk N deposition, precipitation amount, and Nr concentrations in precipitation. All data were analyzed with SPSS statistical software or R software. N deposition spatial patterns were drawn with ArcGIS software. Statistically significant differences were set at $p$ values < 0.05.

## Data availability
Source data are provided with this paper.

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

## Acknowledgements

This study was jointly funded by the National Natural Science Foundation of China (Grants 42277097 to X.J.L., 42175137 to X.W., 41425007 to X.J.L., 42371324 to L.L., 42001347 to L.L.), the High-level Team Project of China Agricultural University to X.J.L., and the Deutsche Forschungsgemeinschaft (DFG)-328017493/GRK 2366 (Sino-German IRTG AMAIZE-P) to X.J.L.

## Author contributions

X.J.L. and F.S.Z. devised the research. Z.W., X.W., L.L., K.W., X.J.L., and K.G. prepared the manuscript. Z.W., X.M., X.W., T.R.S., A.H.T., Ying Zhang, and J.L.S. performed monitoring of atmospheric Nr concentrations for 10 years. A.H.T. and Y.Y.Z. conducted observations of PM$_{2.5}$ concentrations during BRS. M.R.M. and Yu Zhao simulated the variations in N deposition and PM$_{2.5}$ concentrations under climate change by WRF-EMEP model. Y.H.Z. and L.Z. provided Nr deposition velocity by GEOS-Chem model. Z.W. analyzed PM$_{2.5}$ and N deposition trends and assembled meteorological, soil, NDVI, and emission inventory data for machine learning with the help of X.M., X.W., L.L., and X.J.L.

## Competing interests

The authors declare no competing interests.
