## [Peer Review File · Nature Communications]

Combined short-term and long-term emission controls improve air quality sustainably in ChinaREVIEWER COMMENTS

Reviewer #1 (Remarks to the Author):

This article seeks to investigate the favorable outcomes resulting from the integration of short-term and long-term pollution control measures for improving air quality. Through the formulation of a baseline scenario and two additional scenarios, the study quantifies the impact of short-term and long-term emission reductions on PM_{2.5} concentrations and Nr deposition. The findings suggest that both short-term measures and long-term policies prove effective in enhancing air quality.

Several issues necessitate further detailed argumentation and supplementation by the author:

1. Firstly, while this paper quantifies the impacts of short-term measures and long-term policies on PM_{2.5} concentrations and Nr deposition through scenario simulations, it does not simulate the policy effects of combining the short-term measures and long-term policies. Drawing the conclusion that the combination of short-term and long-term policies can effectively achieve pollutant reduction solely based on the observation that short-term measures or long-term policies contributes to emissions reduction lacks credibility and rigor. Additionally, in quantifying the policy effects, the author overlooks potential interference between short-term measures and long-term policies. For example, the short-term measures highlighted, such as those during the Asia-Pacific Economic Cooperation Summit and Military Parades, occurred after the implementation of long-term measures covered in this paper, specifically, the Atmospheric Ten Actions (2013-2017) and the 3-year Defense Battle (2018-2020). The paper does not differentiate the emission reduction effects influenced by short-term measures from those resulting from long-term policies. Consequently, the author does not calculate the “net effects” of short-term or long-term policies, significantly compromising the reliability of the baseline results in this paper.

The author may contemplate the following suggestions: Firstly, quantifying the cost and efficacy of short-term pollution reduction policies after controlling the impact of long-term policies; secondly, assessing the effectiveness of long-term policies on pollution reduction after isolating the influence of short-term policies; thirdly, evaluating the combined effects of short-term measures and long-term policies on pollution reduction; fourthly, conducting a comparative analysis of the implementation costs and effect of these three types of policies, and ultimately deriving conclusions.

2. This paper reviews certain policies and short-term measures aimed at long-term reduction of pollutant emissions. However, there seems to be a lack of specificity in addressing targeted policies for the two pollutants of primary concern in this study, namely PM_{2.5} and reactive nitrogen (Nr) pollution. Could there be consideration for enhancing the specificity in the review of policies?

3. This paper summarizes the significance of exploring Nr pollution but does not identify the reasons of investigating the trends in PM_{2.5}. Please provide additional information.

4. Lines 128-129: There is a lack of evidence to demonstrate that control over SO₂ has been achieved through end-of-pipe treatment rather than source control. Please provide additional information.

5. Lines 165-166: Please provide scientific justification for dividing the time into three stages, namely 2011-2014, 2015-2017, and 2018-2020.

6. Lines 179-182: The argumentation process seems somewhat absolute. Long-term emission control

policies consistently exert pressure on polluting entities, making it less likely to observe pollution rebound after short-term emission control measures conclude. However, this does not necessarily imply that policy implementation has achieved a win-win situation for both environmental protection and economic development. The author is encouraged to further elaborate on how long-term emission control measures can indeed achieve this dual benefit of environmental protection and economic development.

7.Lines 233-236: The citation of references 27-29 appears to lack a clear connection to the discussed content in this paragraph. The focus of this paragraph is primarily on urban and rural areas. Could there be consideration for introducing a discussion on the divergence in marginal abatement costs of NH₃ emissions between urban and rural areas?

8.Lines 283-285: The statement "A recent study ... occurred region" seems somewhat unrelated to the central focus of this paragraph on Nr emissions. Please provide further clarification. Additionally, please review the coherence between the referenced literature and the arguments presented in this paper, ensuring a more logical connection in the reasoning process."

9.Please further review the units of various pollutants in the figures of this paper to ensure accuracy, for instance, "ug N m⁻³" in Figure 4.

10.How short-term measures are specifically integrated with long-term policies needs further clarification. The author is encouraged to provide additional details on this aspect.

11.Should regional heterogeneity be considered when combining short-term and long-term control policies, given the significant differences in economic development stages, industrial structures, energy consumption patterns, and geographical climatic conditions across various regions in China?

12.When formulating pollution control policies, enhancing efficiency and equity is crucial. Please elaborate on how the combination of short-term and long-term control policies reflects efficiency and fairness in the implementation process.

13.Due to the limited feasibility of implementing uniform emission standards for pollutants across different regions, controlling regional air pollution requires not only command-and-control policies but also the integration of economic incentive policies, which aims to harmonize interests among various regions and ensure the full internalization of negative externalities from pollution emissions and positive externalities from pollution control. Please provide additional details on how to establish a sustainable mechanism for pollution control among regions.

Reviewer #2 (Remarks to the Author):

This manuscript provides interesting results. I recommend it be accepted for publication after my following comments are addressed.

Line 28-30. Why are PM_{2.5} and nitrogen trends compared here when PM_{2.5} appears to be presented as results only for August, while nitrogen corresponds to the annual trend? Besides, considering the author presents the reduction rate, it would be beneficial to specify the temporal range rather than using the ambiguous statement "for decades."

Line 34. What does "Nr emissions" refer to? The text lacks an abbreviation note for "Nr," and earlier, the abbreviation for nitrogen was noted as "N." It seems the author may be using "Nr" to represent nitrogen,

but clarification is needed for consistency.

Line 47. What does the author indicate by "People's experience of air pollution is widespread, direct and intense"?

Line 57. "national government" ? I recommend using the "Chinese government" directly.

Line 69-70. Why can the "N deposition" be used to "indicate the direction of air quality improvement"?

Line 75-76. What is the relationship between Nr emission (or N deposition) and climate change? The author does not mention this above.

Line 106. What is "BR"?

Line 112. What is "moral incentive"? I'm not sure about the message the author intends to convey with this sentence.

Line 114-120. The current discussions are not suitable or necessary, as the main text should center around the actual results obtained from the data. Content unrelated to the data of this study should not be addressed here; instead, it could be incorporated into the concluding remarks at the end.

Figure 1. The definition of the periods before and after a specific event are inconsistent in terms of time length, why? Besides, a note of "Table S3" should be placed here for better understanding.

Figure 1e. Why does the author only present the August data here? How about the annual trend?

Line 145. What does the "(NO₃⁻, NH₄⁺)" mean in "2.86 ± 0.39 mg N L⁻¹ (NO₃⁻, NH₄⁺)", unit? And what does the other "N" refer to in the unit in this paragraph?

Line 146-149. What is the difference between "site average" and "geographic annual mean"?

Line 162-166. What's the point of particularly dividing the 10-year period into three stages?

Line 171-173. How is this conclusion reached? The impact of long-term emission control is deemed equally significant as short-term actions.

Line 207-211. How is this conclusion reached? How does the author define "NH₃-rich atmospheric environment"? why "especially for non-point source emissions"? There is no specific statement about the contribution of non-point source emissions here.

Line 255-257. How is this conclusion reached? Why "the increase in NH₃+pNH₄⁺ concentrations from 2018 may be attributed to acid gas emission reductions, despite a reduction shown in the NH₃ emission inventory for this region."

Line 274-276. How is this conclusion reached? How does the author calculate the "transboundary export of pollution"?

Line 296. Why "improvements in air quality could be negated by climate change"? Following the author's reasoning, with climate change leading to increased precipitation, the concentration of pollutants in the air should theoretically decrease. If this holds true, could climate change potentially contribute to improved air quality?

Reviewer #3 (Remarks to the Author):

This manuscript aimed to provide options for short-term and long-term emission controls in China. This is important to sustainably reduce atmospheric pollutant emissions. However, several aspects should be explained more carefully before the paper can be considered for publication. Some suggestions are

offered below:

General comments:

1. It is logical that short-term emergency control measures can quickly improve air quality, while long-term policies take longer to take effect, but are more sustainable. how does your analysis differ from others? This manuscript should present novel findings in this aspect. The analysis of the deposition trends of N_r and NH_3 appears to be novel, more discussions should be paid on the effect of short and long term controls in China.
2. In this manuscript, discussions on short-term controls mostly focused on North China. Other regions like Yangtze River Delta, Fenwen Plain should be added, because there were a lot of short-term controls (for example 2016 G20 Hangzhou summit, and many Regional Air Quality Warnings).
3. I am puzzled about why the author chose to discuss $\text{PM}_{2.5}$ in summer instead of autumn or winter (Figure 1e). After all, China's control measures are more focused on winter, and the most significant effects are also seen in winter. The author should separately discuss the responses of $\text{PM}_{2.5}$ and N deposition in autumn and winter to short-term early warning control and long-term policies.

specific comments:

1. Line 88-101 the difference between pre-, during and post Olympic, APEC et al, is difficult to describe the effect of short-term measures. This because that the meteorology is likely different, which is very important to short-term air quality.
2. Figure 1, Why did the NH_3 concentrations seem no difference during Olympics, APEC and BRs compared to post-APEC and BRs. In urban areas, NH_3 largely came from mobiles and industries, which were reduced during these campaigns.
3. Is the land use changed at sites in Table S1 since 2010-2017? It is key to calculate the deposition velocity in the Formula 1 and analyze its trends.
4. Line 141, this manuscript discussed the long-term trends of N deposition. The short effects are suggested to be added. Discussing only the concentration of $\text{PM}_{2.5}$ and its precursors is not enough to fully evaluate the short-term impact. Many other studies have done so.

Dear Editor:

Please find below our itemized responses to the reviewer's comments. We have addressed the comments raised by three reviewers and incorporated their comments/suggestions in the revised manuscript.

Thank you very much for your consideration.

Sincerely,

Xuejun Liu

Reviewer #1 (Remarks to the Author):

This article seeks to investigate the favorable outcomes resulting from the integration of short-term and long-term pollution control measures for improving air quality. Through the formulation of a baseline scenario and two additional scenarios, the study quantifies the impact of short-term and long-term emission reductions on PM_{2.5} concentrations and Nr deposition. The findings suggest that both short-term measures and long-term policies prove effective in enhancing air quality. Several issues necessitate further detailed argumentation and supplementation by the author:

[Response]: The authors thank the reviewer for the valuable comments and suggestions that have improved our paper. Below we provide a point-by-point response to the reviewer's comments, together with proposed changes in the revised manuscript (in blue).

1. Firstly, while this paper quantifies the impacts of short-term measures and long-term policies on PM_{2.5} concentrations and Nr deposition through scenario simulations, it does not simulate the policy effects of combining the short-term measures and long-term policies. Drawing the conclusion that the combination of short-term and long-term policies can effectively achieve pollutant reduction solely based on the observation that short-term measures or long-term policies contributes to emissions reduction lacks credibility and rigor. Additionally, in quantifying the policy effects, the author overlooks potential interference between short-term measures and

long-term policies. For example, the short-term measures highlighted, such as those during the Asia-Pacific Economic Cooperation Summit and Military Parades, occurred after the implementation of long-term measures covered in this paper, specifically, the Atmospheric Ten Actions (2013-2017) and the 3-year Defense Battle (2018-2020). The paper does not differentiate the emission reduction effects influenced by short-term measures from those resulting from long-term policies. Consequently, the author does not calculate the “net effects” of short-term or long-term policies, significantly compromising the reliability of the baseline results in this paper.

The author may contemplate the following suggestions: Firstly, quantifying the cost and efficacy of short-term pollution reduction policies after controlling the impact of long-term policies; secondly, assessing the effectiveness of long-term policies on pollution reduction after isolating the influence of short-term policies; thirdly, evaluating the combined effects of short-term measures and long-term policies on pollution reduction; fourthly, conducting a comparative analysis of the implementation costs and effect of these three types of policies, and ultimately deriving conclusions.

[Response]: Thank you for this valuable suggestion. Yes, we agree that quantifying the net effect of long-term policies and short-term measures is a more important conclusion of the article. We now specifically discuss the interaction between short-term measures and long-term policies. Firstly, in Res-Figure 1, we present three scenarios with a high level of consensus: **Scenario 1**, assumes no emission reduction measures (and assuming emissions do not rise), where pollutant emissions and concentrations remain stable. **Scenario 2**, considers only short-term measures, in which pollutant emissions and concentrations experience a brief decline followed by a rapid rebound. **Scenario 3**, considering only long-term policies, results in pollutant emissions and concentrations continuing to decline linearly or non-linearly over time.

The impact of short-term measures on long-term policies could be as follows (Res-Figure 2): Hypothesis 1, short-term measures do not enhance the effectiveness of long-term emission reductions. Pollutant emissions and concentrations decline sharply

during the event period, followed by a rapid return to initial levels, and continue to decline linearly or non-linearly with long-term mitigation policy until the reduction target is achieved at time T2. Hypothesis 2, short-term measures will facilitate long-term reductions. The occurrence of the event accelerates the decrease in pollutant emissions and concentrations and the emission reduction target is achieved at an earlier period (T1). In this study, we observed that concentrations of both PM_{2.5} and its precursors almost immediately rebounded to 100% of their pre-event levels after the suspension of the short-term measures. Therefore, we inferred that short-term measures have not significantly altered the slope of emission reductions. Through the Difference-in-Differences (DID) method, based on monthly scales, we further quantified the effects of short-term measures on pollutant concentrations and deposition fluxes. The results showed $p=0.685>0.05$ for PM_{2.5} concentration, $p=0.197>0.05$ for oxidized N deposition, and $p=0.325>0.05$ for reduced N deposition, reaffirming that the trend in long-term measures is not influenced by short-term emission reduction efforts. This could be attributed to the limited scope of emission reductions (Nr emissions from the Beijing-Tianjin-Hebei region account for only 7.5% of the national total emissions) and the short duration of such measures (at most 1 month), which results in the negligible impact of these short-lived reductions on the long-term trend.

We agree that long-term policies may partially overshadow the benefits of short-term measures. Assuming a daily decrease in Nr emissions without taking into account the meteorological factors or transport effects, theoretically, the pollution concentrations of each month should be lower than that of the previous month. Therefore, PM_{2.5} and its precursor concentrations, following short-term measures, should be slightly lower than those before emission reduction. In this study, the difference in PM_{2.5} concentration during the Per-parade and Parade event was 68%, which is significantly higher than the average monthly PM_{2.5} concentration reduction rate of 0.7% in 2015, suggesting that the net effect of the short-term measures is approximately 67%. Similarly, the net effect of the short-term measures during the APEC event was about 61%.

We have included the following discussion: “Short-term measures were virtually unaffected by long-term emission reduction, with short-term net effects of 67% and 61% for APEC and Parade events, respectively. Short-term measures also do not inherently alter the slope of the decline in long-term pollution reduction ($p>0.05$). This is because the decrease in concentrations from long-term policies is stable and gradual. The $>50\%$ reduction in $PM_{2.5}$ concentration achieved in Beijing through one month of short-term measures has in fact taken nearly a decade to stabilize. With continuous adjustments in energy and industrial structures, the concentration of atmospheric pollutants is expected to continue to decline. When the concentration of pollutants is extremely low, it shows that long-term emission reduction policies may have played an extremely significant role, and so the implementation of short-term measures might be ineffective.”.

Res-Figure 1 Effects of long-term and short-term measures on pollutant emissions and concentrations under three scenarios

Res-Figure 2 Two hypotheses on the impact of short-term measures on long-term emission reduction trends

2. This paper reviews certain policies and short-term measures aimed at the long-term reduction of pollutant emissions. However, there seems to be a lack of specificity in addressing targeted policies for the two pollutants of primary concern in this study, namely PM_{2.5} and reactive nitrogen (Nr) pollution. Could there be consideration for enhancing the specificity in the review of policies?

[Response]: We have inserted the target pollutants addressed by the measures and policies in the paper. Including: "...including the suspension of industrial activities to decrease SO₂ and NO_x emissions, and the reduction of vehicle movements in and around the event locations mainly for NO_x...", "such as the suspension of manufacturing facilities and vehicle restrictions for SO₂, NO_x and PM_{2.5} emission...".

Table S1 Control measures implemented in Beijing-Tianjin-Hebei during the Olympic Games, APEC, Military Parade, and BRS

Provinces		Control measures
Olympic Games (Aug. 8–24, 2008)	Beijing	1 Strict and provisional driving restrictions on motor vehicles by odd-even number (for NO _x and PM _{2.5});
		2 Suspended the use of 70% of government motor vehicles (for NO _x and PM _{2.5});
		3 Stopped some construction site operations (for SO ₂ and PM _{2.5});
		4 More effective road cleaning (for PM _{2.5});
		5 Stopped production and limited production in key polluting enterprises (for SO ₂ and NO _x);
		6 Emission pollutants (SO ₂ , NO _x , etc.) decreased by 30% from fuel combustion (for SO ₂ and NO _x);
		7 Organic emissions at gas stations, tankers, and storage depots reduced (for PM _{2.5});
	Tianjin	1 Strict and provisional driving restrictions on motor vehicles by odd-even number (for NO _x and PM _{2.5});
		2 Some gas stations in key areas stopped refueling operations during key hours (for NO _x and PM _{2.5});
		3 Open burning of straw strictly prohibited (for PM _{2.5} and NH ₃);
		4 Non-electricity coal-fired facilities and small thermal power plants ceased operation (for SO ₂ and NO _x);

		5	Selected pollutant emissions suspended (for SO ₂ and NO _x);
		6	Increased road flushing and sprinkling for dust suppression (for PM _{2.5});
	Hebei	1	All production activities related to construction works are suspended (for PM _{2.5});
		2	Increase in the frequency of road water spraying (for PM _{2.5});
		3	Strict control of straw burning and coal burning (for PM _{2.5} , NO _x , SO ₂ , and NH ₃);
		4	Strengthened the inspection of vehicles entering Beijing (for NO _x and PM _{2.5}).
	Beijing	1	Strict supervision and driving restrictions on motor vehicles (for NO _x and PM _{2.5});
		2	Driving ban on 70% of vehicles belonging to government and state-owned enterprises (for NO _x and PM _{2.5});
		3	All construction sites within the fifth ring road suspended (for PM _{2.5});
		4	Production by coal-fired and industrial companies suspended (for SO ₂ and NO _x);
		5	Emission standard of pollutants (SO ₂ , NO _x , etc.) increased by 50% (for SO ₂ and NO _x);
		6	Key roads required to be cleaned at a high frequency every day (for PM _{2.5});
		7	Strict control of express deliveries in Beijing (for PM _{2.5});
APEC (Nov. 1– 12, 2014)	Tianjin	1	50% of motor vehicles limited and public transit capacity enhanced (for NO _x and PM _{2.5});
		2	All production activities related to construction works suspended (for PM _{2.5});
		3	Emission standard of pollutants (SO ₂ , NO _x , etc.) increased by 30% (for SO ₂ and NO _x);
		4	Main roads in the central area cleaned once a day (for PM _{2.5});
		5	Strict control of straw burning and coal burning (for PM _{2.5} and NH ₃);
	Hebei	1	Strengthened inspection of vehicles entering Beijing (for NO _x and PM _{2.5});

		2	Real-time monitoring of straw burning and (for PM _{2.5} and NH ₃);
		3	Emission limits for pollutants increased by 50% in Shijiazhuang (for SO ₂ , NO _x and PM _{2.5});
		4	Construction sites and key enterprises suspended or restricted in key areas (for SO ₂ , NO _x and PM _{2.5}).
	Beijing	1	Strict and provisional driving restrictions on motor vehicles by odd-even number (for NO _x and PM _{2.5});
		2	Driving ban on 80% of vehicles belonging to government and state-owned enterprises (for NO _x and PM _{2.5});
		3	Widespread surprise inspections and rectification of polluting enterprises (for SO ₂ , NO _x and PM _{2.5});
		4	All construction works within the fifth ring road suspended (for PM _{2.5});
		5	Removed from use 217,000 old vehicles and 244 heavy polluting enterprises in advance (for SO ₂ , NO _x and PM _{2.5});
		6	58 important roads cleaned at high frequency every day (for PM _{2.5});
		7	Enhanced supervision of thermal power plants and steel companies in west and south Beijing based on scientific research (for SO ₂ , NO _x and PM _{2.5});
Military Parade (Sep. 3, 2015)	Tianjin	1	Increased emission reduction standard of coal power plants by more than 50% (for SO ₂);
		2	Completed the yellow label car elimination task six months in advance (for NO _x and PM _{2.5});
		3	Strengthened efforts to rectify environmental pollution incidents (for SO ₂ , NO _x and PM _{2.5});
		4	Cars that did not meet national emission standards were not allowed to enter Beijing (for NO _x and PM _{2.5});
		5	Main roads in the central area and Binhai area required to be cleaned once a day (for PM _{2.5});
	Hebei	1	Set up an inspection team to account for environmental violations (for SO ₂ , NO _x and PM _{2.5});
		2	Driving restrictions on motor vehicles by odd-even number (for NO _x and PM _{2.5});
		3	Production of key companies suspended in Baoding and other cities (for SO ₂ , NO _x and PM _{2.5});

		4	Enhanced the desulfurization facilities in ANSTEEL in Handan (for SO ₂ , NO _x and PM _{2.5});
		5	All construction works and pollutant companies suspended or restricted (for SO ₂ , NO _x and PM _{2.5}).
Belt and Road (May. 8–17, 2017)	Beijing	1	The principle of “non-disturbance of the resident”;
		2	Provisional traffic restrictions on vehicles with dangerous chemicals (for NO _x and PM _{2.5});
		3	Some logistics vehicles are completely banned (for NO _x and PM _{2.5});
	4	Enterprises that could not meet emissions standards or with excessive emissions suspended (for SO ₂ , NO _x and PM _{2.5});	
	5	Set up a special inspection team to ensure air quality (for SO ₂ , NO _x and PM _{2.5});	
	Tianjin	1	Strengthened control of coal pollution, industrial pollution and dust pollution (for SO ₂ , NO _x and PM _{2.5});
		2	Strengthened the assessment and accountability mechanism;
	Hebei	1	10-day corporate inspection (for SO ₂ , NO _x and PM _{2.5});
		2	Strengthened environmental law enforcement and data monitoring (for SO ₂ , NO _x and PM _{2.5}).

3. This paper summarizes the significance of exploring Nr pollution but does not identify the reasons of investigating the trends in PM_{2.5}. Please provide additional information.

[Response]: Thank you for pointing this out. We note in the Introduction: “PM_{2.5} concentrations can characterize air pollution variation at a daily scale, responding rapidly to changes in strong emission or meteorological conditions. Atmospheric N deposition, as a sink for Nr emissions, can be used as an indicator to evaluate the degree of Nr pollution at the monthly/yearly scale.”.

4. Lines 128-129: There is a lack of evidence to demonstrate that control over SO₂ has been achieved through end-of-pipe treatment rather than source control. Please provide additional information.

[Response]: We revised the text of “end-of-pipe” to “strengthen industrial emission standards, upgrades on industrial boilers, and phase out outdated industrial capacity¹”.

5. Lines 165-166: Please provide scientific justification for dividing the time into three stages, namely 2011-2014, 2015-2017, and 2018-2020.

[Response]: The division of the past decade into three stages is based on the different development phases of atmospheric environmental issues. The period from 2011 to 2014 was marked by severe air pollution, which gradually attracted the attention of the public and the government. However, during this stage, emission reduction measures were limited. Between 2015 and 2017, air pollution control was of high priority, and the implementation of the “Air Pollution Prevention and Control Action Plan” resulted in the strict control of pollutant emissions, strengthened adjustment of the energy structure, technological transformations in industries, and the enhancement of joint prevention and control mechanisms. From 2018 to 2020, air pollution campaigns achieved remarkable improvements. During this stage, air quality has improved significantly, although the stabilization of these positive trends remains a challenge. In the revised manuscript, we now state that “2011-2014 (period of severe air pollution), 2015-2017 (period of air pollution mitigation), and 2018-2020 (period of air quality significant improvement)”.

6. Lines 179-182: The argumentation process seems somewhat absolute. Long-term emission control policies consistently exert pressure on polluting entities, making it less likely to observe pollution rebound after short-term emission control measures conclude. However, this does not necessarily imply that policy implementation has achieved a win-win situation for both environmental protection and economic development. The author is encouraged to further elaborate on how long-term emission control measures can indeed achieve this dual benefit of environmental protection and economic development.

[Response]: Agreed. We have removed the sentence and added a discussion of environmental protection and economic benefits in the response to Question #12 and Question #13.

7. Lines 233-236: The citation of references 27-29 appears to lack a clear connection to the discussed content in this paragraph. The focus of this paragraph is primarily on urban and rural areas. Could there be consideration for introducing a discussion on the

divergence in marginal abatement costs of NH₃ emissions between urban and rural areas?

[Response]: This is a very good question. Current research on urban NH₃ focuses on source quantification based on isotope techniques, on-site monitoring, emission inventories, etc., but there are few studies on the differences in the cost of NH₃ emission abatement for urban and rural areas. We believe this is a good idea that can be addressed in our future research. Here we add some text: “According to Gu et al. (2023)², the cost of technological measures to reduce global agricultural NH₃ emissions by 32% is around 19 billion USD. However, the technologies used do not take into account the difficulty of technology diffusion and acceptability for farmers, so the actual cost of reducing NH₃ emissions from agricultural sources may be higher than this. Very few studies have been addressed urban NH₃ emission reduction actions. Nonetheless, with adjustments in urban industrial structures and the promotion of electric vehicles and other measures that are mainly focused on NO_x and other pollutants, it is possible to achieve a “free ride” on reducing urban NH₃ emissions, which could significantly lower the economic cost of urban NH₃ emission reduction.”.

8. Lines 283-285: The statement “A recent study ... occurred region” seems somewhat unrelated to the central focus of this paragraph on Nr emissions. Please provide further clarification. Additionally, please review the coherence between the referenced literature and the arguments presented in this paper, ensuring a more logical connection in the reasoning process.”

[Response]: We have added the clarification that: “Pollution transport not only affects N inputs to ecosystems in the surrounding area but also harms human health.”.

9. Please further review the units of various pollutants in the figures of this paper to ensure accuracy, for instance, “ug N m⁻³” in Figure 4.

[Response]: Corrected.

10. How short-term measures are specifically integrated with long-term policies needs further clarification. The author is encouraged to provide additional details on this aspect.

[Response]: Combining this question with Question #12, we have now discussed how

short-term policies can be integrated with long-term strategies in our manuscript. Furthermore, we have added additional content: “Compared to SO₂ and NO₂, it is almost impossible to achieve rapid reductions in NH₃ concentrations quickly. Our case study during the COVID-lockdown period has proven that, after a drastic reduction of transport and industrial NH₃ emissions in February 2020, ambient NH₃ concentrations from January to February (17%) still exceeded the levels of previous years (9%), while concentrations of SO₂ and NO₂ decreased by 24% and 49%, respectively³. This indicates that, as opposed to the intense reductions of acid gases, NH₃ emissions can only be gradually reduced through better fertilizer and manure management. Due to the lack of a national NH₃ monitoring system overseen by the government, the impact of NH₃ on air quality and the environment remains a matter of theoretical debate⁴. Currently, “Green Demonstration Zones” have been established across China, where systemic transformations in precision and smart agriculture will improve nitrogen use efficiency and reduce NH₃ volatilization. We suggest that research on NH₃, PM_{2.5} and atmospheric N deposition should be uniformly strengthened in this area, treating spatial “regional areas” analogously to temporal “short-term” studies, thereby identifying the national potential and environmental effects of NH₃ emission reduction through case studies.”.

11. Should regional heterogeneity be considered when combining short-term and long-term control policies, given the significant differences in economic development stages, industrial structures, energy consumption patterns, and geographical climatic conditions across various regions in China?

[Response]: We agree that there are differences in economic development, energy structures, industrial levels, and meteorological and soil conditions between regions. In the light of these differences, the government has taken them into account when formulating policies or applying emission reduction technologies. From the perspective of combining short-term and long-term controls, there is no directional difference in technology or policy within regions, but only between regions. Within a region, short-term measures aim for immediate effects (such as the implementation of odd-even license plate restrictions), while long-term emission reductions should seek

a socially, economically and environmentally sustainable path (such as promoting electric and hybrid vehicles, enhancing fuel quality and encouraging urban planning that reduces the need for private vehicle use). Besides, for regions experiencing little or no air pollution, short-term emission reduction measures might not yield significant outcomes. However, for areas that suffer serious pollution, a combination of short-term and long-term strategies proves to be more effective.

12. When formulating pollution control policies, enhancing efficiency and equity is crucial. Please elaborate on how the combination of short-term and long-term control policies reflects efficiency and fairness in the implementation process.

[Response]: Thank you for this suggestion. We have added text to the discussion: “Frequent or prolonged implementation of high-intensity short-term control measures will disrupt the development of industry, transport and the residential life. Therefore, at a time when the effectiveness of long-term emission control is still tentative, it is recommended that a short but major emission control exercise be carried out once every 1-2 years to clarify the regional emission reduction potential. Policymakers should gradually adjust the intensity of the controls, and align them with long-term goals, ensuring a smooth transition and continuous progress towards objectives, defining specific and achievable long-term targets for air quality improvements that are informed by the success of short-term reductions. In the long-term emission reduction process, two aspects need to be clarified. Firstly, “polluter identification”. For example, sintering in the steel industry is a major source of NO_x emissions, while in agriculture, high amounts of NH₃ are emitted from the treatment of manure from livestock. Therefore, to increase pollution management efficiency, the specific targets of emission reduction must be identified (including key emitting sectors and critical emission stages). Secondly, the principle of “polluter pays” should be achieved. To ensure fairness in environmental and economic terms within pollution management, it is essential to identify the input and output of pollutants between regions. There needs to be a more precise quantification of transboundary contributions, which can provide data support for subsequent compensatory mechanisms between regions.”.

13. Due to the limited feasibility of implementing uniform emission standards for

pollutants across different regions, controlling regional air pollution requires not only command-and-control policies but also the integration of economic incentive policies, which aims to harmonize interests among various regions and ensure the full internalization of negative externalities from pollution emissions and positive externalities from pollution control. Please provide additional details on how to establish a sustainable mechanism for pollution control among regions.

[Response]: Thank you for this insightful comment. We have added relevant text to the discussion: “Quantifying the economic benefits and environmental costs associated with pollution transport contributes to revealing the environmental inequalities of regional emissions and pollution. As reflected in this study, economically developed regions export Nr emissions to neighbouring regions, which results in these suffering both economic and environmental losses. Currently, economic compensation between regions under multiple constraints (air pollution, climate change, ecological effects) is hardly discussed. Regions more economically developed are encouraged to increase green investments in emission-reducing technological transformations⁵ and may need to compensate polluted regions around them. This could be through economic measures such as tax incentives, subsidies and loans, encouraging enterprises and consumers to make more environmentally friendly choices. For instance, financial support could be provided to enterprises and farms that utilize clean energy and energy-efficient technologies, and tax relief is given to consumers for purchasing environmentally friendly products and services.”.

Reviewer #2 (Remarks to the Author):

This manuscript provides interesting results. I recommend it be accepted for publication after my following comments are addressed.

[Response]: We thank the reviewer for recommending publication. Below we provide a point-by-point response to the reviewer’s comments, together with proposed changes in the revised manuscript (in blue).

1. Line 28-30. Why are PM2.5 and nitrogen trends compared here when PM2.5

appears to be presented as results only for August, while nitrogen corresponds to the annual trend? Besides, considering the author presents the reduction rate, it would be beneficial to specify the temporal range rather than using the ambiguous statement "for decades."

[Response]: In combination with reviewer #3's Question #3, we have answered this by adding more details about winter variations (Figure 1f) of precursor, PM_{2.5}, and SNA concentrations for the period of 2005-2020 in the revised paper: "Although PM_{2.5} concentrations in Beijing are significantly higher in winter than in summer, they still showed a continuous downward trend over the 15 years: from January 2005 to January 2020, PM_{2.5}, SNA, SO₂ and NO₂ decreased by 65%, 51%, 94% and 50%, respectively. In contrast to the summer months, NH₃ concentrations increased by about 50% in winter and stabilized at 7 µg m⁻³ after 2013.", and "Similarly, the air pollution episode in January 2013 caused strong dissatisfaction among the government and public of Beijing. However, from that time until 2020, the PM_{2.5} concentration in Beijing decreased by approximately 58%, ...". In the Abstract, we clarify this: "Long-term adherence to sustained and strict emission reduction policies led to successful decreases of 58% in PM_{2.5} concentrations in Beijing, and of 23% in atmospheric nitrogen (N) deposition in China during the period of 2011-2020...".

Figure 1 PM_{2.5}, gaseous precursors (SO₂, NO₂, NH₃) and secondary inorganic ion (SNA: SO₄²⁻, NO₃⁻, NH₄⁺) concentrations (average \pm SD) in Beijing during (a) pre-Olympic Games (Jun.1-30, 2008), Olympic Games (Aug.8-24, 2008) and post-Olympic Games (Sep.24-Oct.23, 2008); (b) pre-APEC (Oct.1-31, 2014), APEC (Nov.1-12, 2014) and post-APEC (Nov.13-Dec.31, 2014); (c) pre-Military Parades (Aug.3-19, 2015), Military Parades (Aug.20-Sep.3, 2015) and post-Military Parades (Sep.3-29, 2015); (d) pre-Belt and Road Summit (Apr.28-May 7, 2017), Belt and

Road Summit (May 8-17, 2017) and post- Belt and Road Summit (May 18-Jun.17, 2017 May 18-Jun.17, 2017); (e) the period of August 2005 to August 2020; (f) the period of January 2005 to January 2020.

2. Line 34. What does "Nr emissions" refer to? The text lacks an abbreviation note for "Nr," and earlier, the abbreviation for nitrogen was noted as "N." It seems the author may be using "Nr" to represent nitrogen, but clarification is needed for consistency.

[Response]: Here Nr emissions refer to emissions from various reactive N compounds (i.e., NH₃ and NO_x). We have added an abbreviation note for Nr in the Abstract at line 39 in the revised paper.

3. Line 47. What does the author indicate by "People's experience of air pollution is widespread, direct and intense"?

[Response]: It means that air pollution is common and serious giving those who breathe polluted air a strong uncomfortable feeling. This is widely experienced by most Chinese people, so we need to formulate a series of policies and implement measures to mitigate air pollution.

4. Line 57. "national government"? I recommend using the "Chinese government" directly.

[Response]: Agreed and done.

5. Line 69-70. Why can the "N deposition" be used to "indicate the direction of air quality improvement"?

[Response]: The trend in N deposition can be used to indicate the direction of air quality improvement. This is because atmospheric Nr (reactive nitrogen) components are constituents of PM_{2.5} (such as NH₄⁺ and NO₃⁻ in fine particles) as well as direct components of Nr in air (NH₃, NO₂ and HNO₃). N deposition also contains information about the variation of Nr concentrations in precipitation. If N deposition decreases, it means that the atmospheric Nr concentrations have decreased, so that the PM_{2.5} concentration, which is closely related to air quality, decreases accordingly. Combined with reviewer #1's comments on Question #3, we now explain this in the introduction: "PM_{2.5} concentrations can characterize air pollution variations at a daily scale, responding rapidly to changes in strong emissions or meteorological conditions.

Atmospheric N deposition, as a sink for Nr emissions, can be used as an indicator to evaluate the degree of Nr pollution at the monthly/yearly scale.”.

6. Line 75-76. What is the relationship between Nr emission (or N deposition) and climate change? The author does not mention this above.

[Response]: Thank you for pointing this out. We now state in the Introduction “In addition, global warming may contribute to changes in spatial patterns and interannual trends in atmospheric Nr emissions and N deposition due to the high correlation between temperature and NH₃ emissions.”.

7. Line 106. What is “BR”?

[Response]: We apologize for this mistake, it should be BRS, representing the Belt and Road Summit. This is now explained in the text (line 117).

8. Line 112. What is “moral incentive”? I'm not sure about the message the author intends to convey with this sentence.

[Response]: The sentence has been revised as “This means that, in order to improve air quality standards in the short term, it is necessary to adopt some unconventional emission reduction measures at the expense of the economy to remind the public that pollution can be prevented and so increase their confidence in policies and practices.”.

9. Line 114-120. The current discussions are not suitable or necessary, as the main text should center around the actual results obtained from the data. Content unrelated to the data of this study should not be addressed here; instead, it could be incorporated into the concluding remarks at the end.

[Response]: Agreed and revised accordingly. These sentences have been incorporated into the concluding remarks.

10. Figure 1. The definition of the periods before and after a specific event are inconsistent in terms of time length, why? Besides, a note of “Table S3” should be placed here for better understanding.

[Response]: For Olympic, APEC and parade events, the data are from the literature. The three periods are categorized as pre-event (one month before the event), event, and post-event (one month after the event). For the Belt and Road Summit event, since no significant emission reduction measures were implemented, our sampling

frequency at that time was 1 time per day for both pre-event and event, and once 3 days after the event. This ensured that there were 10 PM_{2.5} samples available for analysis during the period of pre-BRS, BRS and post-BRS. For a clearer explanation of this, in Figure 1, we labeled the duration of each event. Please also see our response to Question #1.

11. Figure 1e. Why does the author only present the August data here? How about the annual trend?

[Response]: See our previous answer and that to Question #1. We have added the January trend in the revised paper.

12. Line 145. What does the “(NO₃⁻, NH₄⁺)” mean in “2.86 ± 0.39 mg N L⁻¹ (NO₃⁻, NH₄⁺)”, unit? And what does the other “N” refer to in the unit in this paragraph?

[Response]: The N_r in the air (including 5 N_r species: NH₃, NO₂, HNO₃, pNH₄⁺, pNO₃⁻) and N_r in precipitation (including 2 N_r species: NO₃⁻, NH₄⁺) are not the same. The mean concentrations of N_r in precipitation (2.86 ± 0.39 mg N L⁻¹) are in different units to those N species in the air (units = μg N m⁻³). However, expressing these in terms of N rather than NO₃⁻ or NO₂ allows the different dry and wet deposition component fluxes to be added together, after they have been calculated, based on deposition velocity and rainfall amount as kg N ha⁻¹ yr⁻¹.

13. Line 146-149. What is the difference between “site average” and “geographic annual mean”?

[Response]: The “site average” is the mean value of atmospheric N deposition at 59 monitoring sites. These are not evenly distributed across the region. The number of monitoring sites is higher in the North China Plain and Northeast China (more populous, more economically developed, and more polluted) than in other regions, and the lower number of sites in some regions is constrained by electric power and labor. Using “site averages” may overestimate the true level of N deposition across China. Therefore, we calculated the “geographic mean”, which is the average N deposition in each region multiplied by the proportion of the area that each region accounts for the entire country, summing the means. This reduces the overestimation of the true N deposition flux and “site averages”. The relevant equations are as

follows, now included in Materials and Methods.

$$F = F_d + F_b$$

Site average N deposition

$$= (F_{NC1} + F_{NC2} + \dots + F_{SE1} + F_{SE2} + \dots + F_{SW1} + F_{SW2} + \dots + F_{NW1} + F_{NW2} + \dots + F_{NE1} + F_{NE2} + \dots + F_{TP1} + F_{TP2} + \dots)/59$$

Geographic average N deposition

$$= \left[(F_{NC1} + F_{NC2} + \dots) / \left(\frac{A_{NC}}{960} \right) \right] + \left[(F_{SE1} + F_{SE2} + \dots) / \left(\frac{A_{SE}}{960} \right) \right] + \left[(F_{SW1} + F_{SW2} + \dots) / \left(\frac{A_{SW}}{960} \right) \right] + \left[(F_{NW1} + F_{NW2} + \dots) / \left(\frac{A_{NW}}{960} \right) \right] + \left[(F_{NE1} + F_{NE2} + \dots) / \left(\frac{A_{NE}}{960} \right) \right] + \left[(F_{TP1} + F_{TP2} + \dots) / \left(\frac{A_{TP}}{960} \right) \right]$$

F_{NCi} represents total N deposition at site NCi, and A_i represents the area of that particular region i.

14. Line 162-166. What's the point of particularly dividing the 10 years into three stages?

[Response]: Please see our response to Reviewer #1's Question #5.

15. Line 171-173. How is this conclusion reached? The impact of long-term emission control is deemed equally significant as short-term actions.

[Response]: We want to emphasize the greater effectiveness of long-term policies compared to the quick rebound of pollutant concentrations under short-term measures. To avoid confusion, we have revised the text as: "Compared to immediate, effective but short-lived short-term pollution control measures, long-term policies take more time to have an impact, but their effects are more permanent."

16. Line 207-211. How is this conclusion reached? How does the author define "NH₃-rich atmospheric environment"? why "especially for non-point source emissions"? There is no specific statement about the contribution of non-point source emissions here.

[Response]: Thank you for pointing it out. The "NH₃-rich atmospheric environment" here represents a relatively high concentration of NH₃ in the atmosphere. We added the categorization of "NH₃-rich" or "NH₃-poor" to further quantify the degree of "NH₃-richness"⁶. The formula is as follows:

$$AdjGR = \frac{\text{free ammonia}}{\text{total nitrate}} = \frac{TA - DSN \times TS}{TN} = \frac{[NH_3](g) + [NO_3^-](p)}{[NO_3^-](p) + [HNO_3](g)}$$

$$DSN = \frac{[NH_4^+](p) - [NO_3^-](p)}{[SO_4^{2-}](p)}$$

$$TA = [NH_3](g) + [NH_4^+](p)$$

$$TN = [HNO_3](g) + [NO_3^-](p)$$

$$TS = [SO_2](g) + [SO_4^{2-}](p)$$

The concentration unit of gaseous and particulate pollutants is $\mu\text{mol m}^{-3}$. Where *AdjGR* of <1 indicates an NH_3 -poor regime, indicating that small perturbations in NH_3 emissions would have a significant effect on SIAs. Likewise, an *AdjGR* of >1 indicates NH_3 -rich conditions that have a free NH_3 ratio capable of neutralizing excess HNO_3 produced by the additional increase in NO_x emission. The national average value of *AdjGR* is 3.24, which is more than 1, so we consider that our country is generally NH_3 -rich atmospheric environment.

17. Line 255-257. How is this conclusion reached? Why “the increase in $\text{NH}_3 + \text{pNH}_4^+$ concentrations from 2018 may be attributed to acid gas emission reductions, despite a reduction shown in the NH_3 emission inventory for this region.”

[Response]: The increase of NH_3 concentrations in the atmosphere can be attributed to two main factors. On the one hand, it may be due to a direct increase in NH_3 source emissions. On the other hand, it may be a consequence of reduced emissions of acid precursors (SO_2 , NO_x), leading to fewer neutralization reactions with NH_3 and so more remaining of gaseous NH_3 in the air. Both scenarios are reasonable contributors to rising atmospheric NH_3 levels. In southwest China, emission inventories have shown a decreasing trend in atmospheric NH_3 emissions (MEIC, declined by 10% from 2017 to 2020). However, there has been an observed increase in NH_x concentrations. Consequently, it is concluded that the reduction in NH_3 emissions is relatively minor and insufficient to offset the concentration increase caused by the decreased acidic gas emissions (SO_2 emission declined by 36%, NO_x emission declined by 11%, and the decrease in observed oxidized N concentrations in this study is also evidence of a reduction in acidic gas emissions). Similarly, Liu et al. (2021)⁷ found that the considerable decrease of atmospheric nitric acid concentrations in

Southern China resulted in an increase of gaseous NH₃ concentrations by 8.0% and consequently enhanced NH₃ dry deposition.

To avoid confusion, we now state that “In contrast, the NH₃ emission inventory for the Southwest region shows a 10% decrease from 2017-2020. Therefore, an observed increase in atmospheric NH₃+pNH₄⁺ concentrations may be attributed to a decrease in acid gas emissions (SO₂ emission declined by 36%, NO_x emission decline by 11%) and fewer neutralization reactions, resulting in more NH₃ present in the atmosphere.”.

18. Line 274-276. How is this conclusion reached? How does the author calculate the “transboundary export of pollution”?

[Response]: For the quantification of pollution transport between regions, we assume that Nr emissions and deposition in China are balanced, with no pollutants being transported to China from other countries. The emission-to-deposition process between regions is simplified: all pollutants will preferentially be deposited in their region and *Tran* is the difference between the provincial Nr emissions and N deposition. If *Tran* is positive, Nr emissions > N deposition and the residual pollutants will be transported to other regions. If *Tran* is negative, Nr emissions < N deposition and all pollutants are deposited in their region. We have added a description of this assumption to Methods.

$$Tran = Nr\ emission - N\ deposition$$

19. Line 296. Why “improvements in air quality could be negated by climate change”? Following the author's reasoning, with climate change leading to increased precipitation, the concentration of pollutants in the air should theoretically decrease. If this holds true, could climate change potentially contribute to improved air quality?

[Response]: The relationship between air quality and climate change is not simple. First, climate change includes extreme precipitation, consisting of droughts and storms. Focusing on precipitation, heavy rainfall contributes to the removal of pollutants from the air, whereas droughts may lead to an increase in pollutant concentrations. On the other hand, the reduction in the concentration of particulates/aerosols in the atmosphere is followed by more solar radiation being captured by the surface (reduced reflectivity)⁸. The accelerated emission due to the

warming effect is probably an important contributor to the increased pollution. Overall, substantial emission reductions are needed to offset the effects of global warming.

Reviewer #3 (Remarks to the Author):

This manuscript aimed to provide options for short-term and long-term emission controls in China. This is important to reduce atmospheric pollutant emissions sustainably. However, several aspects should be explained more carefully before the paper can be considered for publication. Some suggestions are offered below:

[Response]: We thank the reviewer for the comments. Below we provide a point-by-point response to the reviewer's comments and proposed changes in the revised manuscript (in blue).

General comments:

1. It is logical that short-term emergency control measures can quickly improve air quality, while long-term policies take longer to take effect, but are more sustainable. how does your analysis differ from others? This manuscript should present novel findings in this aspect. The analysis of the deposition trends of Nr and NH₃ appears to be novel, more discussions should be paid on the effect of short and long-term controls in China.

[Response]: Most previous studies have demonstrated either the effectiveness of short-term pollution control measures or long-term emission reduction policies. However, the issue of pollution rebound from short-term measures, and the contribution of short-term measures to long-term emission reductions has been little discussed. Firstly, in the short term, where the improvement of air pollution is temporary and the economic costs are extremely high, is it necessary to implement short-term abatement measures only to safeguard national image? Secondly, most studies on the effectiveness of long-term policies concentrate on trends in PM_{2.5} and precursor concentrations. In addition to the much-discussed PM_{2.5} pollution, we have included N deposition as an indicator, which includes air quality information as

concentrations of gaseous, particulate Nr in air, and Nr in precipitation, and the ecological impacts of surface N inputs via atmospheric deposition. This provides a more comprehensive understanding of the future challenges that need to be addressed for better air quality and eco-environmental protection.

Combining our responses to Reviewer #3 with those of Reviewer #1 and Reviewer #2's suggestions, we have added relevant details to highlight the novelty of the article.

First, we clarify the research objectives of this article in the Introduction, i.e., “To ensure “blue skies”...”, “In order to sustain “blue skies”, the Chinese government has issued a series of policies aimed at long-term reductions in pollutant emissions (i.e, that should persist for years) ...”, “...and uncovers hidden challenges in creating effective improvements in air quality”, “PM_{2.5} concentrations can characterize air pollution variations at a daily scale, responding rapidly to changes in strong emission or meteorological conditions. Atmospheric N deposition, as a sink for Nr emissions, can be used as an indicator to evaluate the degree of Nr pollution at the monthly/yearly scale.”.

Second, we further quantify the net effect of short-term and long-term emission reductions (please see response to Reviewer #1, Question #1). In brief, our study reveals that short-term measures have negligible effects on the slope of long-term emission reductions based on a Difference in Differences (DID) analysis. Short-term measures are more important in demonstrating the potential for emission reduction and enhancing people's confidence in the likely effectiveness of long-term emission reductions. Furthermore, short-term measures were virtually unaffected by long-term emission reduction. Additionally, we have expanded the discussion on how short-term measures and long-term policies can be integrated, covering the coupling of emission reduction targets, the synergy between different precursors, and the economic costs of regional transfers, to highlight the theme of the paper.

Third, short-term measures have an insignificant impact on N deposition on a monthly scale, as N deposition comprises 7 Nr species and is the combination of the deposition velocity and Nr concentrations in gaseous form, and the rainfall amount and Nr concentrations in precipitation. Consequently, our article focuses on the day-

scale variations in PM_{2.5}, SNA, and precursor concentrations, as well as the annual trends in N deposition.

2. In this manuscript, discussions on short-term controls mostly focused on North China. Other regions like Yangtze River Delta, Fenwen Plain should be added, because there were a lot of short-term controls (for example 2016 G20 Hangzhou summit, and many Regional Air Quality Warnings).

[Response]: Thank you for this suggestion. Due to the limitations of the monitoring data, we were unable to cover all short-term emission reductions, but we have added some text to the discussion: “Similarly, a decrease of 12% in PM_{2.5} and 12% in SNA concentrations was observed for the Hangzhou G20 Summit compared to the period before aggressive mitigation measures were introduced. However, after the various control measures were lifted, all air pollutant concentrations increased back to initial levels⁹. During the 2022 Beijing Winter Olympics, the intense emission controls significantly reduced PM_{2.5} and SNA concentrations, with source control measures contributing 54% to the total reduction in PM_{2.5} concentrations¹⁰.”.

3. I am puzzled about why the author chose to discuss PM_{2.5} in summer instead of autumn or winter (Figure 1e). After all, China's control measures are more focused on winter, and the most significant effects are also seen in winter. The author should separately discuss the responses of PM_{2.5} and N deposition in autumn and winter to short-term early warning control and long-term policies.

[Response]: Combining with Reviewer #2's suggestions, we have added a description of winter PM_{2.5} pollution, please see our response to Reviewer #2's Question #1.

We also added the text in the revised paper: “In contrast to PM_{2.5} concentrations, monthly averaged atmospheric N deposition was significantly higher in summer than in winter by 32% (Figure S1). However, similar to PM_{2.5} concentrations, under the influence of long-term emission reductions, the alleviation of summer pollution is greater than that in winter. Winter air quality clearly needs to be improved.”.

Figure S1 Average monthly values of atmospheric N deposition in winter and summer from 2011 to 2020.

specific comments:

1. Line 88-101 the difference between pre-, during and post Olympic, APEC et al, is difficult to describe the effect of short-term measures. This because that the meteorology is likely different, which is very important to short-term air quality.

[Response]: Thank you for pointing this out. We have added the relevant discussion: “Meteorological conditions need to be taken into account to determine the actual impact of the control measures on pollution mitigation, which is important for short-term case studies. Liu et al.¹¹ showed that emission control measures during the APEC reduced PM_{2.5} concentrations in Beijing by an average of 41%, and by more than 50 $\mu\text{g m}^{-3}$ under unfavorable meteorological conditions. At the time of the Parade and Pre-Parade, the integrated analyses of modeling and monitoring demonstrate that pollutant emission reduction measures are the main contributor to this as there were no significant differences in the two periods between the temperature, wind speed, and relative humidity¹².”.

2. Figure 1, Why did the NH₃ concentrations seems no difference during Olympics, APEC and BRs compared to post-APEC and BRs. In urban areas, NH₃ largely came from mobiles and industries, which were reduced during these campaigns.

[Response]: Partly correct. During the event, the concentration of atmospheric NH₃ did not show a significant decreasing or increasing trend, whereas the amount of

aerosol NH_4^+ was reduced by more than 50%, and the overall NH_x concentration decreased, reflecting the impact of emissions source changes on NH_3 . The insignificant variation in atmospheric NH_3 can be attributed to two causes. Firstly, the background level of atmospheric NH_3 emissions is relatively high in the North China Plain region, which has extensive agricultural activities. Compared to acidic gases, there is an excess of atmospheric NH_3 . Even though some NH_3 has been converted to aerosol NH_4^+ through reactions with acidic gases, a large amount of NH_3 still exists in the atmosphere in gaseous form, resulting in stable atmospheric NH_3 concentrations. Secondly, most events such as Olympics, Parade, BRS and APEC took place in warm months, and most studies indicate that agriculture remains the main source of NH_3 in Beijing during the summer. Isotopic analysis suggests that about 30%-40% of atmospheric NH_3 in summer originates from agriculture, and about 20%-30% from sewage treatment^{13,14}. Therefore, even if transportation and industrial activities were suspended during an event, there would still be the other dominant sources of NH_3 . Our previous research indicates that during the COVID-2019 lockdown periods³, the concentration of atmospheric NH_3 in Beijing in winter still increased, highlighting the importance of transported agricultural and human sources for urban NH_3 .

3. Is the landuse changed at sites in Table S1 since 2010-2017? It is key to calculate the deposition velocity in the Formula 1 and analyze its trends.

[Response]: Land use types at sites in Table S1 did not change from 2011-2020. The GEOS (Goddard Earth Observing System)-Chem chemical transport model (CTM; <http://geos-chem.org>) was used to simulate the V_d values of five Nr species (gases NH_3 , NO_2 , HNO_3 , and aerosols $p\text{NH}_4^+$, $p\text{NO}_3^-$) every hour at each monitoring site from 2011 to 2020. Hourly V_d values were averaged to obtain a monthly V_d , which was multiplied by the monthly species concentration to estimate the dry deposition flux. The V_d values of gaseous NH_3 , NO_2 , HNO_3 and particulate NH_4^+ and NO_3^- all showed no significant differences from 2011 to 2020 ($p>0.05$) (Figure S8). Therefore, calculated trends in dry deposition are not a function of V_d , but follow the measured Nr concentrations in air.

Figure S8 Variations in the dry deposition velocities of Nr species (a: NH₃, b: HNO₃, c: NO₂, d: pNH₄⁺, e: pNO₃⁻) at sites within the National Nitrogen Deposition Monitoring Network (NNDMN) from 2011 to 2020.

4. Line 141, this manuscript discussed the long-term trends of N deposition. The short effects are suggested to be added. Discussing only the concentration of PM_{2.5} and its precursors is not enough to fully evaluate the short-term impact. Many other studies have done so.

[Response]: Thank you for this suggestion. Please see our response to Question #1 where we explain our response and our enhanced discussion of short-term measures.

Reference

1. Zhang Q, *et al.* Drivers of improved PM_{2.5} air quality in China from 2013 to 2017. *Proceedings of the National Academy of Sciences of the United States of America* **116**, 24463-24469 (2019).
2. Gu, B. *et al.* Cost-effective mitigation of nitrogen pollution from global croplands. *Nature* **613**, 77-84 (2023).
3. Xu W, *et al.* Increasing importance of ammonia emission abatement in PM_{2.5} pollution control. *Science bulletin* **67**, 1745-1749 (2022).
4. Liu M, *et al.* Ammonia emission control in China would mitigate haze pollution and nitrogen deposition, but worsen acid rain. *Proceedings of the National Academy of Sciences of the United States of America* **116**, 7760 (2019).
5. Cao S, Nie L, Sun H, Sun W, Taghizadeh-Hesary F. Digital finance, green technological innovation and energy-environmental performance: Evidence from China's regional economies.

Journal of Cleaner Production **327**, 129458 (2021).

6. Pinder RW, Dennis RL, Bhawe PV. Observable indicators of the sensitivity of PM_{2.5} nitrate to emission reductions—Part I: Derivation of the adjusted gas ratio and applicability at regulatory-relevant time scales. *Atmospheric Environment* **42**, 1275-1286 (2008).
7. Liu M, *et al.* Unexpected response of nitrogen deposition to nitrogen oxide controls and implications for land carbon sink. *Nature Communications* **13**, 3126 (2022).
8. Wang P, *et al.* Aerosols overtake greenhouse gases causing a warmer climate and more weather extremes toward carbon neutrality. *Nature Communications* **14**, 7257 (2023).
9. Ji Y, *et al.* Counteractive effects of regional transport and emission control on the formation of fine particles: a case study during the Hangzhou G20 summit. *Atmospheric Chemistry and Physics* **18**, 13581-13600 (2018).
10. Liu Y, Xu X, Yang X, He J, Ji D, Wang Y. Significant Reduction in Fine Particulate Matter in Beijing during 2022 Beijing Winter Olympics. *Environmental Science & Technology Letters* **9**, 822-828 (2022).
11. Liu H, *et al.* The blue skies in Beijing during APEC 2014: A quantitative assessment of emission control efficiency and meteorological influence. *Atmospheric Environment* **167**, 235-244 (2017).
12. Xu W, *et al.* Air quality improvement in a megacity: implications from 2015 Beijing Parade Blue pollution control actions. *Atmospheric Chemistry and Physics* **17**, 31-46 (2017).
13. Li Y, *et al.* Apportioning Atmospheric Ammonia Sources across Spatial and Seasonal Scales by Their Isotopic Fingerprint. *Environmental Science & Technology* **57**, 16424-16434 (2023).
14. Zhang Y, Benedict KB, Tang A, Sun Y, Fang Y, Liu X. Persistent Nonagricultural and Periodic Agricultural Emissions Dominate Sources of Ammonia in Urban Beijing: Evidence from ¹⁵N Stable Isotope in Vertical Profiles. *Environmental Science & Technology* **54**, 102-109 (2020).

REVIEWER COMMENTS

Reviewer #1 (Remarks to the Author):

1.The paper aims to enhance the credibility of the effectiveness of short-term and long-term emission reduction strategies. To further quantify the impact of short-term measures on pollutant concentrations, the author employs the Difference-in-Differences (DID) method. To bolster the credibility of the conclusions drawn from the DID model and demonstrate that long-term measures remain unaffected by short-term emission reduction efforts, it is essential to present the constructed DID model. This should encompass details on the treatment group, control group, selection of control variables, chosen exogenous shocks related to short-term measures, and a discussion on potential endogeneity issues. Please elaborate further.

2.Figure 5: Is this the complete version of the national map?

3.Lines 409-412: It mentions that current air quality control measures, focused on end-of-pipe treatment and clean production, are conducive to achieving air pollution emission control targets but carry the risk of suppressing reductions in CO₂ emissions. I don't quite understand why air quality control measures could pose a risk of suppressing reductions in CO₂ emissions. The author cites two references here, and based on Reference 41, the conclusion is that the contribution of low-carbon policies to improving air quality is greater than end-of-pipe control. Additionally, Reference 41 mentions that since CO₂ as an air pollutant shares sources with emissions, climate policies can reduce pollutant emissions, improve air quality, and simultaneously reduce carbon emissions. It seems that this does not lead to the conclusion of "the risk of suppressing reductions in CO₂ emissions". Please explain.

Reviewer #2 (Remarks to the Author):

The authors have addressed my comments well.

Reviewer #3 (Remarks to the Author):

The revised manuscript responded to most of my previous concerns, and seems to have made significant improvements. However, Minor comments are recommended before the publication.

1. The paper investigates the impact of meteorological fields on short-term PM_{2.5} through citations of prior research. Nevertheless, I believe that this examination is inadequate. studies to distinguish between short-term controlling measures and meteorological fields should be fully discussed.

2. This manuscript employed the GEOS-Chem to estimate the dry deposition and revealed the absence of any notable patterns.The authors note that the land use representation in GEOS-Chem remains unchanged, contradicting real situation. I proposed at least a brief discussion on the impact of land use changes over the past ten years.

Dear Editor:

Please find below our itemized responses to the reviewer's comments. We have addressed all comments raised by three reviewers and incorporated their comments/suggestions in the revised manuscript.

Thank you very much for your consideration.

Sincerely,

Xuejun Liu

Reviewer #1 (Remarks to the Author):

1. The paper aims to enhance the credibility of the effectiveness of short-term and long-term emission reduction strategies. To further quantify the impact of short-term measures on pollutant concentrations, the author employs the Difference-in-Differences (DID) method. To bolster the credibility of the conclusions drawn from the DID model and demonstrate that long-term measures remain unaffected by short-term emission reduction efforts, it is essential to present the constructed DID model. This should encompass details on the treatment group, control group, selection of control variables, chosen exogenous shocks related to short-term measures, and a discussion on potential endogeneity issues. Please elaborate further.

[Response]: We have employed the DID method to quantitatively examine whether and to what extent the short-term emission control measures influence the long-term policies. DID recognizes the NC1(Beijing) as the treatment group, while the other sites are regarded as the control group. Differences for each group in PM_{2.5} concentrations, oxidized N deposition, reduced N deposition during APEC, and Parade (emissions data were lacking of Olympics events, and limited data available after the implementation of short-term measures during the BRS period and therefore did not take part in the assessment) when compared before and after the implementation of the short-term measures are then calculated. According to the result of Random Forest (in the Materials and Methods), the important variables were selected as emission data (SO₂

emission, NO_x emission, NH₃ emission), meteorological parameters (2 m temperature, atmospheric pressure, 10 m wind speed, boundary layer height) and NDVI. The difference in the abovementioned differences thus represents the net effect on PM_{2.5} concentration, oxidized N deposition, and reduced N deposition brought by the short-term measures, and statistically significant differences were set at *p* values < 0.05. The DID method requires there to be a similarity between the treatment group and the control group. In response to this, parallel trends were tested by the event analysis method, with *p* values > 0.05 indicated pass the test. All actions were performed in Stata MP version 18.

2. Figure 5: Is this the complete version of the national map?

[Response]: Thank you for pointing this out. Corrected.

Figure 5 Annual average PM_{2.5}-SNA concentrations and N deposition in China in 2017, 2030 and 2060, using the WRF-EMEP model under the most stringent policy implementations and rising temperatures and increased extreme precipitation (RCP4.5_2030, RCP4.5_2060) in 2030 and 2060.

3. Lines 409-412: It mentions that current air quality control measures, focused on end-of-pipe treatment and clean production, are conducive to achieving air pollution

emission control targets but carry the risk of suppressing reductions in CO₂ emissions. I don't quite understand why air quality control measures could pose a risk of suppressing reductions in CO₂ emissions. The author cites two references here, and based on Reference 41, the conclusion is that the contribution of low-carbon policies to improving air quality is greater than end-of-pipe control. Additionally, Reference 41 mentions that since CO₂ as an air pollutant shares sources with emissions, climate policies can reduce pollutant emissions, improve air quality, and simultaneously reduce carbon emissions. It seems that this does not lead to the conclusion of "the risk of suppressing reductions in CO₂ emissions". Please explain.

[Response]: Previous studies have shown that end-of-pipe treatment has drastically reduced the emissions of air pollutants in the past years, while have not achieved the co-benefits of reduced carbon emissions^{1, 2}. We have revised the sentence from “risks inhibiting CO₂ emission reductions” to “may not be able to reach the synergistic objectives of air quality improvement and climate change mitigation”. Replace reference #41 with “Cheng J, et al. A synergistic approach to air pollution control and carbon neutrality in China can avoid millions of premature deaths annually by 2060. *One Earth* 6, 978-989 (2023)”³.

Reviewer #2 (Remarks to the Author):

The authors have addressed my comments well.

[Response]: We thank the reviewer's recommendation for publication.

Reviewer #3 (Remarks to the Author):

The revised manuscript responded to most of my previous concerns, and seems to have made significant improvements. However, Minor comments are recommended before the publication.

1. The paper investigates the impact of meteorological fields on short-term PM_{2.5} through citations of prior research. Nevertheless, I believe that this examination is

inadequate. Studies to distinguish between short-term controlling measures and meteorological fields should be fully discussed.

[Response]: We have added the analysis and description of four event meteorological parameters, including temperature, precipitation, atmospheric pressure, relative humidity, wind speed, and direction. Now state that “Air pollutant concentrations were mostly significantly negatively linked with wind speed, precipitation, and relative humidity, but have positive correlation with atmospheric pressure. The effect of temperature on air pollution has high uncertainty⁴. In this study, some of the meteorological conditions were favorable for the air pollutants dispersion during the event, such as the occurrence of precipitation during the Parade and stronger northerly winds during the APEC. However, emissions reduction measures remained the most important driver of pollution mitigation. Higher temperatures, elevated relative humidity, lower atmospheric pressure, and frequent southerly winds were ineffective in halting significant decreases in PM_{2.5} concentration during the Olympics compared to the pre- and post-Olympics periods.”.

Figure S1 Temperature (a), precipitation (b), atmospheric pressure (c), relative humidity (d) during the Olympics, APEC, Parade, and BRS.

Figure S2 Wind speed and direction during the Olympics (a), APEC (b), Parade (c), and BRS (d), denoting three subperiods of the pre-event (left), event (middle), and post-event (right), respectively.

2. This manuscript employed the GEOS-Chem to estimate the dry deposition and revealed the absence of any notable patterns. The authors note that the land use representation in GEOS-Chem remains unchanged, contradicting real situation. I proposed at least a brief discussion on the impact of land use changes over the past ten years.

[Response]: The trend variation of dry deposition is primarily influenced by the atmospheric Nr concentration (C_d) and deposition velocity (V_d), as expressed in the following formula:

$$F_d = C_d \times V_d$$

The C_d is obtained through sample collection, measurement and analysis, and V_d is simulated via the GEOS-Chem model. We found C_d decreased by 19% from 2012 to 2020, while V_d exhibited no significant change (see previous response). Consequently, the dry deposition decreased by approximately 19% during this period. Therefore, we infer that this temporal variation in dry deposition is determined by changes in atmospheric Nr concentration.

Stability in land use type and underlying surface is the main reason for the insignificant trend in V_d . For example, at NC9, a rural site in Hebei province, the land use type has always been agricultural. The wheat-maize rotation system has remained stable for many years, although recent improvements in agricultural technology may occurred due to economic development, such as changes in fertilization rates and field management practices.

We have added a brief discussion: “Innovations in industrial or agricultural technology could directly alter the emissions and concentrations of Nr. While the stability of land use types maintains the resistance of gases to capture by surface plants or soil and V_d , thereby emphasizing the key impact of Nr concentration on the trend of dry deposition in this study.”.

Reference

1. Lei Y, *et al.* The 2022 report of synergetic roadmap on carbon neutrality and clean air for China: Accelerating transition in key sectors. *Environmental Science and Ecotechnology* **19**, 100335 (2024).
2. Zhang Q, *et al.* Drivers of improved PM_{2.5} air quality in China from 2013 to 2017. *Proceedings of the National Academy of Sciences of the United States of America* **116**, 24463-24469 (2019).
3. Cheng J, *et al.* A synergistic approach to air pollution control and carbon neutrality in China can avoid millions of premature deaths annually by 2060. *One Earth* **6**, 978-989 (2023).
4. Liu Y, Zhou Y, Lu J. Exploring the relationship between air pollution and meteorological conditions in China under environmental governance. *Scientific Reports* **10**, 14518 (2020).

REVIEWERS' COMMENTS

Reviewer #1 (Remarks to the Author):

The authors have addressed my comments well.